# An acidophilic fungus promotes prey digestion in a carnivorous plant

Pei-Feng Sun [1,2,3], Min R. Lu [1], Yu-Ching Liu [1], Brandon J. P. Shaw [4,5], Chieh-Ping Lin [1], Hung-Wei Chen [1], Yu-fei Lin [1], Daphne Z. Hoh [1], Huei-Mien Ke [6], I-Fan Wang [7,8], Mei-Yeh Jade Lu [1], Erica B. Young [9], Jonathan Millett [4], Roland Kirschner [10], Ying-Chung Jimmy Lin [11,12], Ying-Lan Chen [7,8] & Isheng Jason Tsai [1,2,3] ✉

Leaves of the carnivorous sundew plants (*Drosera* spp.) secrete mucilage that hosts microorganisms, but whether this microbiota contributes to prey digestion is unclear. We identified the acidophilic fungus *Acrodontium crateriforme* as the dominant species in the mucilage microbial communities, thriving in multiple sundew species across the global range. The fungus grows and sporulates on sundew glands as its preferred acidic environment, and its presence in traps increased the prey digestion process. *A. crateriforme* has a reduced genome similar to other symbiotic fungi. During *A. crateriforme–Drosera spatulata* coexistence and digestion of prey insects, transcriptomes revealed significant gene co-option in both partners. Holobiont expression patterns during prey digestion further revealed synergistic effects in several gene families including fungal aspartic and sedolisin peptidases, facilitating prey digestion in leaves, as well as nutrient assimilation and jasmonate signalling pathway expression. This study establishes that botanical carnivory is defined by adaptations involving microbial partners and interspecies interactions.

Botanical carnivory has evolved independently at least 11 times in the plant kingdom, each group showcasing distinct molecular adaptations to attract, trap and digest insects[1]. Many carnivorous plants have served as research models since the era of Charles Darwin[2] to understand the evolutionary and molecular basis of carnivorous structures, which are frequently related to adaptations of leaf organs. These specialised leaves secrete digestive exudates that may also host a diverse array of microorganisms. Although the significance of microorganisms in vertebrate digestion is widely established[3], recent research has suggested

symbiotic roles for microorganisms in carnivorous plants[4], but the underlying molecular responses through which microorganisms facilitate or enhance plant carnivory are only just emerging[4–6].

Plant–microorganism interactions are highly dynamic and can impact plant fitness through many mechanisms[7]. Previous studies using metabarcoding suggested that the digestive mucilage secreted by modified leaves, or traps, of carnivorous plants, can be colonised by diverse bacterial and eukaryotic microbial communities (Extended Data Table 1). In bladderwort, corkscrew and pitcher plants, there were

[1]Biodiversity Research Center, Academia Sinica, Taipei, Taiwan. [2]Biodiversity Program, Taiwan International Graduate Program, Academia Sinica and National Taiwan Normal University, Taipei, Taiwan. [3]Department of Life Science, National Taiwan Normal University, Taipei, Taiwan. [4]Geography and Environment, Loughborough University, Loughborough, UK. [5]NERC Environmental Omics Facility (NEOF), NEOF Visitor Facility, School of Biosciences, University of Sheffield, Sheffield, UK. [6]Department of Microbiology, Soochow University, Taipei, Taiwan. [7]Department of Biotechnology and Bioindustry Sciences, College of Bioscience and Biotechnology, National Cheng Kung University, Tainan, Taiwan. [8]University Center of Bioscience and Biotechnology, National Cheng Kung University, Tainan, Taiwan. [9]Department of Biological Sciences, University of Wisconsin-Milwaukee, Milwaukee, WI, USA. [10]School of Forestry and Resource Conservation, National Taiwan University, Taipei, Taiwan. [11]Department of Life Science, College of Life Science, National Taiwan University, Taipei, Taiwan. [12]Institute of Plant Biology, College of Life Science, National Taiwan University, Taipei, Taiwan. ✉e-mail: ijtsai@gate.sinica.edu.tw

no single dominant species represented, but diverse bacteria can be broadly grouped into major phyla[5,8–12]. Bacterial diversity and biomass were found to improve prey decomposition rates in the pitcher plant *Darlingtonia californica* and increase nitrogen uptake efficiency in host leaves[12]. Meta-transcriptomic profiling of *Genlisea* species revealed non-host transcripts dominated by metazoan hydrolases, suggesting a role in phosphate acquisition[6]. The composition of the microbiota also appears to be highly time dependent and influenced by factors such as host plant[12], community succession[12,13], surrounding environment and prey possibly contributing bacteria[14], so there are complex interactions that shape these microbial communities.

*Drosera* is a genus known as sundews within the Droseraceae, the second largest carnivorous plant family after Lentibulariaceae[15]. Sundews have 'flypaper' leaf traps with tentacle-like trichomes[16] that secrete sticky mucilage to entrap and envelop prey. Subsequent insect digestion is a well-coordinated process, first by synthesis and secretion of digestive enzymes to break down organic materials, followed by uptake and assimilation of nutrients with specialised transporters[17,18]. This complex behaviour is mediated by the jasmonate (JA) signalling pathway that was present in non-carnivorous plant ancestors[19]. Recent comparative genomics of carnivorous plants has revealed expansion and clade-specific gene families involved in defence, such as JA signalling, as well as peptidases and hydrolases[20–22], which were upregulated during the digestion process suggesting that these genes have been co-opted[23] for new roles in carnivory. The extent to which these genes still retain their ancestral functions remains to be elucidated.

We hypothesize that carnivorous plants harbour microorganisms that positively enhance their digestion process, The digestion process therefore involves a plant–microorganism holobiont[24], in which specific microbial taxa would facilitate digestion within carnivorous plants and others may be neutral or even pathogenic to the host plant. To test this hypothesis, we focus on *Drosera spatulata*[25] (Fig. 1a), a sundew native to temperate and tropical regions including Taiwan and which has a fully sequenced genome[21]. The fascinating mechanism of trap movement in sundews, associated with prey capture and digestion, has attracted significant scientific curiosity. The traps also act as a tractable system to examine responses to diverse stimuli[22] in both natural and laboratory settings. We aimed to characterise the sundew mucilage microbial community and assess their impact on digestion of insect prey. We identified *Acrodontium crateriforme* as the dominant fungal species that enhances prey digestion efficiency in *D. spatulata*, showing that its enzymatic activities synergize with the plant's own digestive processes. Our research aims to shed light on the intricate symbiotic interactions between carnivorous plants and their resident microorganisms.

## Results

### Distinct fungal communities in *Drosera* mucilage

The microbial diversity and composition of the mucilage of 92 samples (Fig. 1b and Supplementary Table 1) collected from the sundew *D. spatulata* and surrounding non-carnivorous plants growing on cliff habitats (Supplementary Fig. 1) across three sites in northern Taiwan (Supplementary Fig. 2) and characterised by 16S rRNA and internal transcribed spacer (ITS) amplicon metabarcoding showed diverse bacterial and fungal communities. The average bacterial and fungal operational taxonomic units (OTUs) of the samples were 580 and 604, respectively. The bacterial species diversity was similar between the sundew mucilage and leaf surfaces of co-occurring plants (Fig. 1c; Wilcoxon rank sum test, $P = 1$). By contrast, the fungal communities in *D. spatulata* mucilage had significantly lower Shannon diversity than those in co-occurring plants (Fig. 1c; Wilcoxon rank sum test, $P < 0.001$). The beta diversity of microbial communities indicated significant differences in bacterial (Fig. 1d; PERMANOVA, leaf surface: $R^2 = 0.15$, $P = 0.001$) and fungal (Fig. 1e; PERMANOVA, leaf surface: $R^2 = 0.27$, $P = 0.001$) communities between leaves of *Drosera* and those of other plant sources.

Around 89.6% and 95.1% of bacterial and fungal OTUs corresponding to 72.5–100% relative abundance of mucilage microbiome, respectively, were also found in the co-occurring plant samples (Supplementary Table 2), implying that the microbial composition in mucilage was similar that of the co-occurring non-carnivorous plants, but these taxa differed in their relative abundances.

### Dominance of *A. crateriforme* in *Drosera* plants

The relative abundances of the five most common bacterial and fungal taxa in leaf mucilage revealed a dominant fungal OTU averaging 41.8% sequence relative abundance (Fig. 2a). This dominant fungal OTU was identified as *A. crateriforme*. Monthly sampling of mucilage and ITS sequencing across two sites for 9 months revealed that *A. crateriforme* maintained its status as the most abundant fungal species, despite lower relative abundance (21.7%) during July and August (Fig. 2b). To investigate the extent of *A. crateriforme* dominance in *Drosera*, we carried out ITS sequencing of sundew mucilage and tissue samples collected in Taiwan, the United Kingdom and the United States, which revealed *A. crateriforme* presence in all four *Drosera* species over three continents (Fig. 2c and Extended Data Fig. 1). This suggests that the *Drosera*–*A. crateriforme* coexistence is well conserved and may be ancient. *A. crateriforme* remained the most dominant species in 98.1% of mucilage samples spanning ~44 km of northern Taiwan, with an average relative abundance of 29.4% (Fig. 2c and Extended Data Fig. 1a). Particularly high *A. crateriforme* dominance (>50% relative abundance) was found in *Drosera rotundifolia* tissues sampled in the United Kingdom and the United States, but was less common in UK samples of *Drosera anglica* and *Drosera intermedia* (Fig. 2c). Furthermore, reanalysis of 14 published fungal metabarcoding datasets of carnivorous plants detected the presence of *A. crateriforme* also in the purple pitcher plant *Sarracenia purpurea*[26], although with a low abundance of 0.2–1.4% (Extended Data Table 1). Together, these results suggest varying symbiotic dynamics of *A. crateriforme* across different carnivorous plant species.

We isolated the two most abundant OTUs from the environment of *D. spatulata* as *A. crateriforme* and *Phoma herbarum* of the families Teratosphaeriaceae and Didymellaceae, respectively (Extended Data Fig. 2). *P. herbarum* is known as a plant pathogen causing leaf spot in various crops[27]. Like *D. spatulata*, *A. crateriforme* can be cultivated under laboratory conditions using potato dextrose agar (PDA) and Murashige and Skoog (MS) agar. The acidophilic *A. crateriforme* grows optimally at pH 4–5, a similar pH to *D. spatulata* mucilage[28] (Supplementary Fig. 3a), while *P. herbarum* preferred a neutral pH. The optimal culture temperature for *A. crateriforme* is 25 °C (Supplementary Fig. 3b), aligning with the growth range for *D. spatulata* (7–32 °C) and the average monthly temperature of 22 °C at the sampling sites (Supplementary Fig. 3c). The summer temperature peaks (29–35 °C) at the Taiwanese sites, combined with biotic factors such as optimal growth of *P. herbarum* at this higher temperature range, may explain the reduced abundance of *A. crateriforme* at higher temperature (Fig. 2b).

### Plant–fungus coexistence enhances sundew digestion

Examining the stalk glands of *D. spatulata* growing in sterile lab culture using a scanning electron microscope (SEM) revealed clear uncolonised gland surfaces (Fig. 3a and Extended Data Fig. 3a–c). Inoculating the sundew with *A. crateriforme* resulted in hyphae growing over the glands (Fig. 3b and Extended Data Fig. 3d–f), and conidiophores and detached conidia were observed in glands collected from wild specimens (Fig. 3c and Extended Data Fig. 3g–i). This observation shows that *A. crateriforme* colonises and reproduces on the sundew stalk glands[29], and the mucilage harbours free hyphae or conidiophores. Cultivation of *A. crateriforme* increased when *Polyrhachis dives* ant powder was added to the medium, suggesting that the fungus can use insects as a growth supplement (Supplementary Fig. 4).

To examine the potential contribution of *A. crateriforme* to the sundew host, *A. crateriforme* was inoculated onto the *D. spatulata* leaves

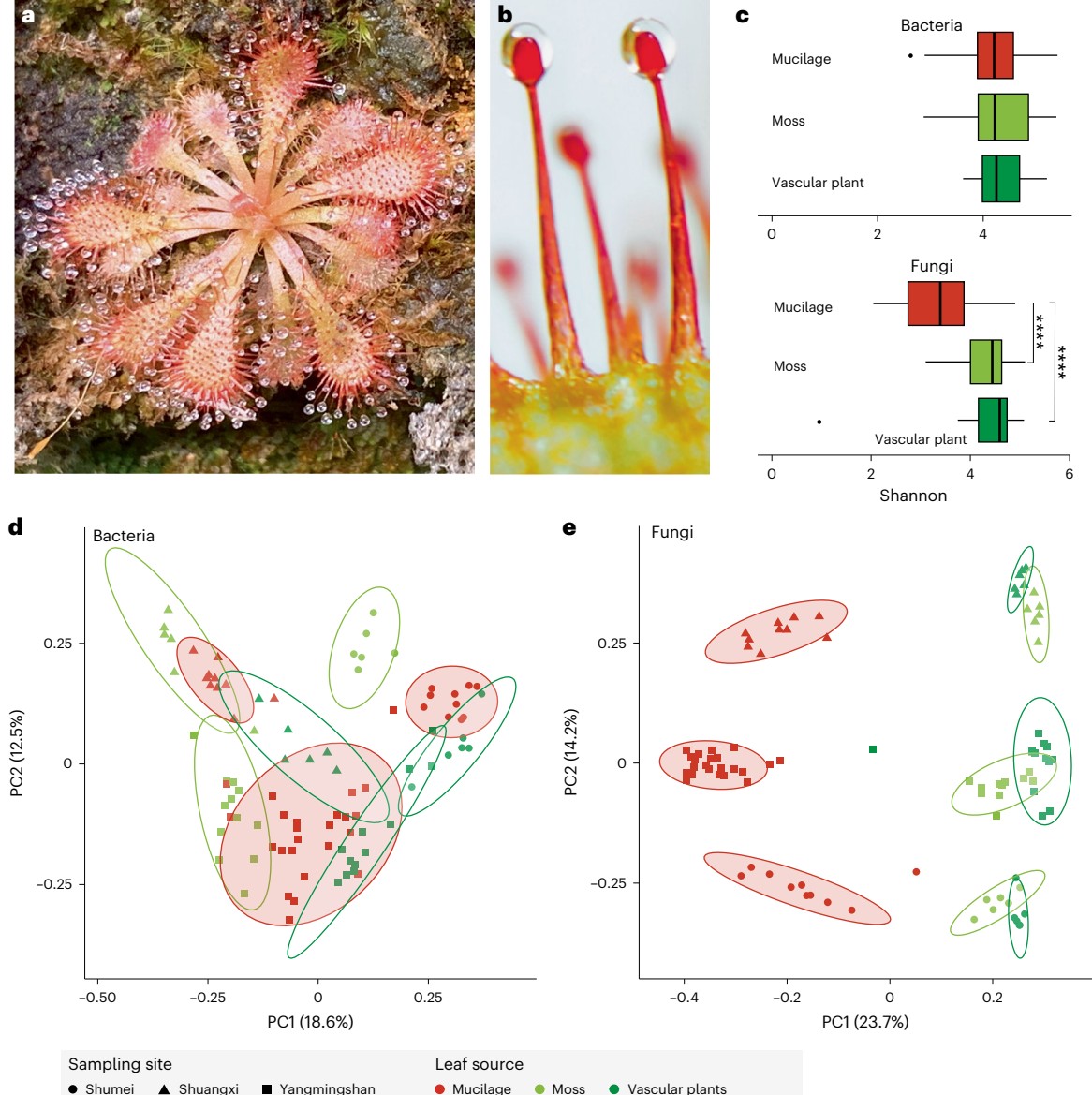

**Fig. 1 | Microbial communities of *D. spatulata* mucilage and surrounding co-occurring plants. a**, *D. spatulata*. **b**, Close-up of stalk glands with secreted mucilage. **c**, Bacterial and fungal species evenness in sundew mucilage (*n* = 44) versus those of co-occurring moss (*n* = 24) and vascular plant leaf surfaces (*n* = 24). Asterisks denote significant difference from Wilcoxon rank sum test (two sided with multiple testing, ****adjusted *P* < 0.0001; mucilage versus vascular plant, *P* = 1.80 × 10⁻⁸; mucilage versus moss, *P* = 1.79 × 10⁻⁷). The centre line represents the median, and the upper and lower bounds of the box represent the 25th and 75th percentiles, respectively. The whiskers extend to 1.5 times the interquartile range (i.q.r.). **d**,**e**, Beta diversity (Bray–Curtis index) of bacterial (**d**) and fungal (**e**) communities from three sites (Supplementary Fig. 2). Ellipses were drawn at 95% confidence level within samples of the same plant source and site. PC1, principal component 1; PC2, principal component 2.

with no significant responses in plant morphology and net weight after 1 month (Extended Data Fig. 4a). By contrast, inoculating *P. herbarum* onto *D. spatulata* leaves at the same concentration resulted in plant wilt (Extended Data Fig. 4b). When the leaves were supplemented with ant powder, the recorded time required for the stalk glands to fully cover the prey, completely digest the ant powder and then return to their original position averaged 92 h (Fig. 3d). Mechanical stimulation or supplementation of sundew leaves with non-natural materials (shrimp and wood powders) resulted in much quicker reopening of traps (-23 h; Extended Data Fig. 5a), suggesting that the plant was able to distinguish between prey providing nutrients and non-prey. Recolonizing the sterile sundew leaves with the following three different microbial communities: (1) *A. crateriforme* only, (2) sundews' natural microbiota and (3) leaf surface microbiota of plants co-occurring with sundews revealed a significantly reduced reopening time of traps when the

microorganisms were present than in sterile leaves (adjusted *P* < 0.01, unpaired two-tailed *t*-test; Fig. 3d). This suggests that the presence of a microbial community enhanced the sundews' responses against substrates. Significantly faster trap reopening was observed in plants inoculated with *A. crateriforme* only than in those inoculated with microbiota of the surrounding environment from co-occurring plants (median, 73 h versus 81 h; unpaired two-tailed *t*-test, adjusted *P* = 0.012; Fig. 3d), showing the fungus's direct and positive involvement in substrate digestion on sundew leaves. The addition of protease inhibitor significantly lengthened the reopening time in all treatments (averaging 182.8 h; Extended Data Fig. 5b), corroborating previous findings that the peptidases were involved in digestion[30–32].

To establish whether microorganisms facilitate prey digestion by increasing protein degradation capability, digestion rates of the applied biotinylated bovine serum albumin (BSA) showed that mucilage from

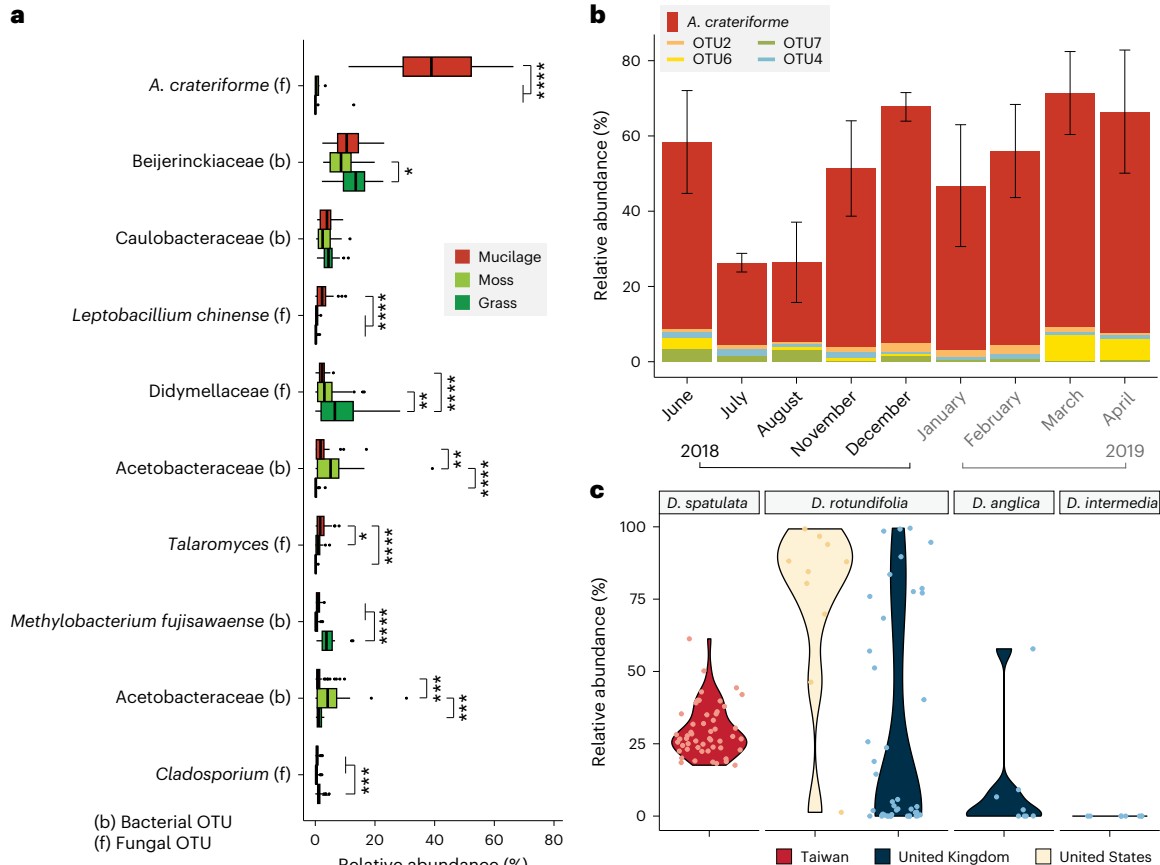

**Fig. 2 | Abundance of *Acrodontium* OTU in *Drosera* mucilage. a**, Relative abundances of the ten most abundant bacterial and fungal taxa among mucilage of *D. spatulata* (*n* = 44) and surrounding plants (moss, *n* = 24, and vascular plants, *n* = 24), highlighting *A. crateriforme* as the dominant species in the mucilage. Unpaired *t*-test was conducted (two sided, *\*P* < 0.05; *\*\*P* < 0.01; *\*\*\*P* < 0.001; *\*\*\*\*P* < 0.0001). The centre line represents the median, and the upper and lower bounds of the box represent the 25th and 75th percentiles, respectively.

The whiskers extend to 1.5 times the i.q.r. **b**, Temporal variation of *Acrodontium* and the next four most abundant OTUs shown as relative abundance in mucilage over 9 months (*n* = 4) between 2018 and 2019 pooled from Shumei and Shuangxi in northern Taiwan (Supplementary Fig. 2). Data are presented as mean ± s.d. **c**, Relative abundance of *A. crateriforme* identified from ITS amplicons from mucilage or tissues of four *Drosera* species sampled across northern Taiwan, North America and the United Kingdom.

recolonised plants actively reduced BSA remaining over time (Supplementary Fig. 5). Significant reduction of BSA after 24 h was observed in mucilage from sundews inoculated with only *A. crateriforme* but not the samples inoculated with natural microbiota (85.9% versus others: 92.4–94.1%, adjusted *P* = 0.012, Wilcoxon rank sum exact test, Fig. 3e; with repeated experiments also showing the same trend in Extended Data Fig. 6). This emphasises that more proteins were being digested with *A. crateriforme* present and that *A. crateriforme* is a functional part of the sundew holobiont enhancing digestion.

**Genome of *A. crateriforme* as an extremophilic fungus**
To examine genetic potential for digestive functions, the full genome of *A. crateriforme* was sequenced and assembled using 10.5 Gb of Oxford Nanopore long reads and consensus sequences polished with Illumina reads. The final assembly resulted in 14 contigs, with 13 containing TTAGGG copies at both ends corresponding to gapless chromosomes (Supplementary Table 3). The assembly size of 23.1 Mb represents the first genome from the genus *Acrodontium*. We predicted 8,030 gene models using the MAKER2 (ref. 33) pipeline aided by transcriptome read mappings as hints. Of these, 97.3% of the predicted gene models were found to be orthologous to at least one of the 25 representative species in the order Capnodiales (Supplementary Table 4), suggesting a conserved core genome with some potentially unique adaptations. A species phylogeny was constructed by coalescing 9,757 orthogroup trees, which placed *A. crateriforme* within a group of extremophilic

species (Fig. 4a) including the well-known acidophilic fungi *Acidomyces richmondensis* and *Neohortaea acidophila*[34]. The mating locus and its adjacent orthologues of *A. crateriforme* were determined, and this was syntenic with sister species (Supplementary Fig. 6), suggesting that this fungus is probably heterothallic, similar to the ancestral Mycosphaerellales[35].

We functionally annotated the *A. crateriforme* proteome to identify genes and gene families associated with metabolism (Fig. 4b). Principal component analysis of protein family domain numbers from each species first differentiated the extremophiles *Friedmanniomyces simplex* and *Hortaea werneckii* from others with their partial[36] or whole[37] duplicated genome (Supplementary Fig. 7a). *A. crateriforme* was positioned between its extremophile relatives and outgroup plant pathogens (Supplementary Fig. 7b). Among the extremophiles, inference of gene family dynamics indicated a relatively high number of losses in *A. crateriforme* similar to the human dermatophyte *Piedraia hortae* (Fig. 4a). *A. crateriforme* has lost members of glycoside hydrolase 6, 11, 28 and 43, which degrade plant cellulose cell walls, and their losses have been implicated as signatures of symbiotic fungi[38] such as ectomycorrhizal fungi[39] (Extended Data Fig. 7 and Supplementary Table 5). Further specializations of *A. crateriforme* include a high number of polyketide synthase (PKS) clusters (Fig. 4b), and most identified PKS biosynthetic gene clusters (BGCs) in *A. crateriforme* were not shared with other representative species, implying that it has a distinct profile of polyketides from secondary metabolite pathways

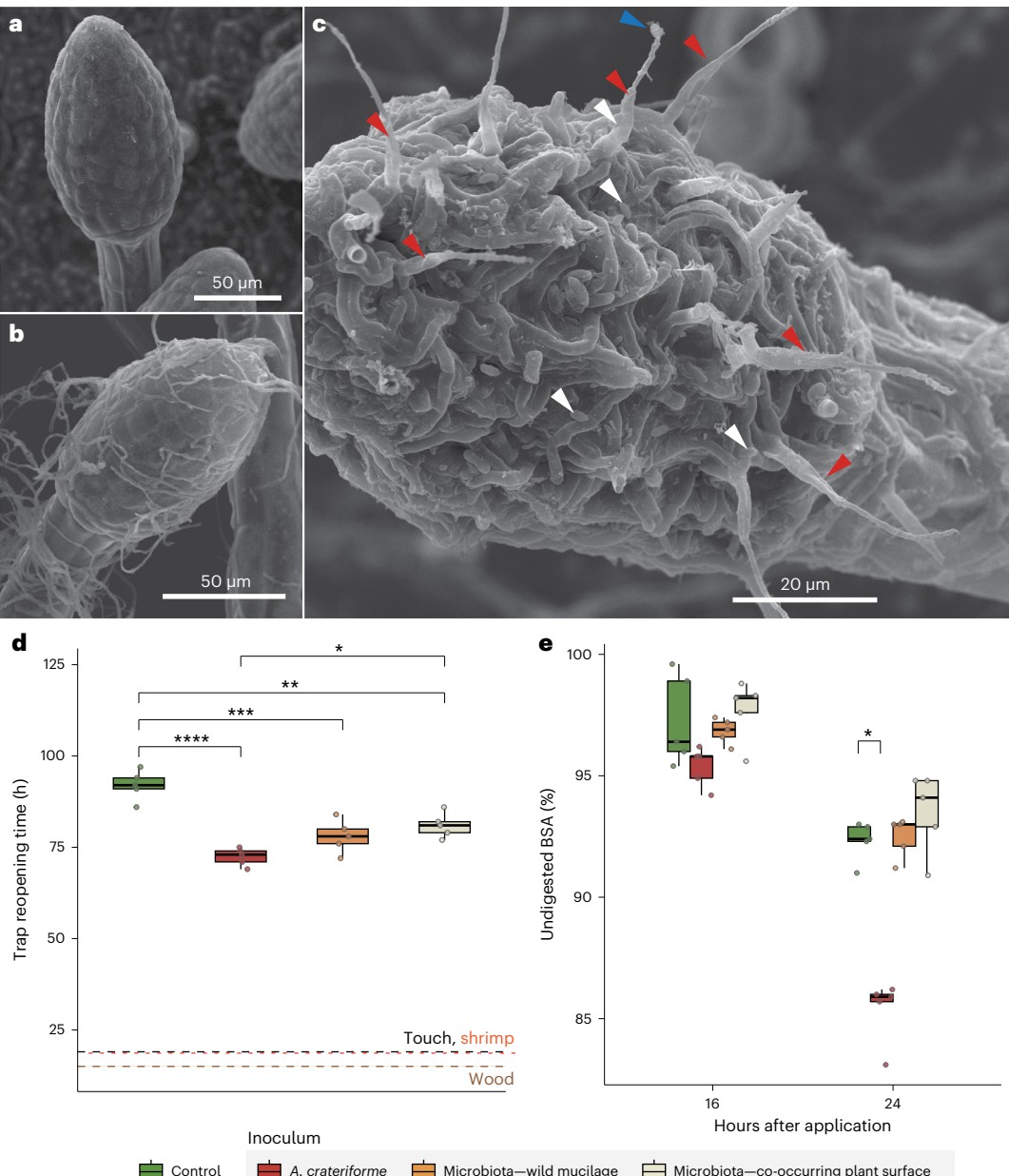

**Fig. 3 | The *A. crateriforme*–*D. spatulata* holobiont. a–c**, SEM image of sundew stalk glands under sterile lab conditions (**a**), inoculated with *A. crateriforme* (**b**) and from wild samples from the natural habitat (**c**). Each condition was repeated once and images were retaken with similar results. Different arrow colours denote fungal conidiophores from which conidia were already detached (red), a conidium attached to the tip of a conidiophore (blue) and detached conidia (white). **d**, Reopening time of sundew traps with and without (control) supplementation with different substrates, or with inoculation with different microbiota. Each set contains a single leaf from five individual plants. The centre line represents the median, and the upper and lower bounds of the box represent the 25th and 75th percentiles, respectively. The whiskers extend to 1.5 times the i.q.r. Median reopening time for touch, shrimp and wood in sundews with different microbiota are plotted as dashed lines (Extended Data Fig. 5). Unpaired two-tailed *t*-tests with multiple testing were performed (*$P < 0.05$; **$P < 0.01$; ***$P < 0.001$; ****$P < 0.0001$). **e**, Application of biotin-labelled BSA as a protein substrate during 16 h and 24 h of sundew digestion showing a decline in BSA with digestion using collected mucilage from sundews inoculated with different inocula. Each set contains five samples of pooled mucilage from a single sundew or entire wiped leaf surfaces of a single adjacent plant. Raw western blot images are shown in Supplementary Fig. 22. The centre line represents the median, and the upper and lower bounds of the box represent the 25th and 75th percentiles, respectively. The whiskers extend to 1.5 times the i.q.r. Wilcoxon rank sum test with multiple testing was conducted (*$P < 0.05$).

(Supplementary Fig. 8). *A. crateriforme* encodes two peptidase Neprosin domain-containing genes (Supplementary Fig. 9), which were absent in all representative species and rare in fungi (149 versus 8,118 in Viridiplantae; InterPro, last assessed October 2023). Neprosin was first discovered in Raffles' pitcher plant *Nepenthes rafflesiana* as a novel peptidase capable of digesting proteins at low concentrations without substrate size restriction[40], functions which may be valuable in prey digestion.

To characterise the mode of genome evolution, we identified on average 4,747 pairwise single-copy orthologues between *A. crateriforme* and sister extremophiles. Clustering of these orthologues with corresponding *A. crateriforme* chromosomes identified only one one-to-one linkage group on chromosome 13 (Fig. 4c), suggesting frequent chromosomal fusions and fissions since their last common ancestor. Gene order within linkage groups has been lost (Supplementary Fig. 10), suggesting extensive intra-chromosomal

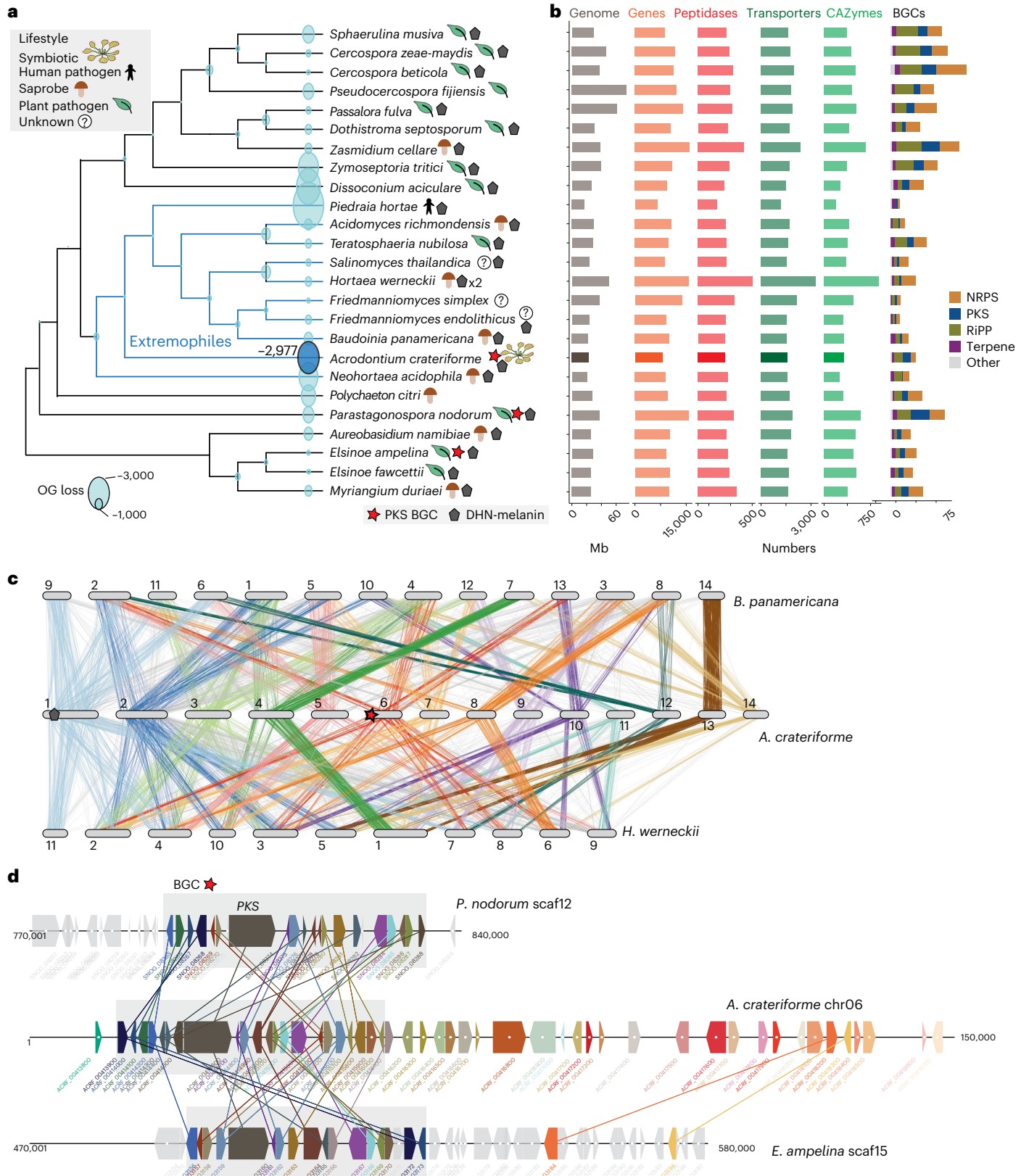

**Fig. 4 | Genomic features of A. crateriforme. a**, Phylogenetic placement of *A. crateriforme* among extremophilic fungi denoted in blue branches, highlighting its association with known acidophiles. All nodes have a 100% bootstrap support. The number next to *A. crateriforme* denotes the number of lost OGs inferred by DOLLOP. **b**, Genome description and functional annotations of the *A. crateriforme* proteome. DHN, dihydroxynaphthalene; NRPs, nonribosomal peptides; RiPP, ribosomally synthesised and post-translationally modified peptides. **c**, Chromosomal rearrangements among extremophile fungi *A. crateriforme*, *Baudoinia panamericana* and *H. werneckii* through clustering of single-copy orthologue pairs. The line colours designate corresponding *A. crateriforme* chromosomes. **d**, Synteny between a subtelomeric polyketide cluster on *A. crateriforme* chromosome 6 and plant pathogens *E. ampelina* and *P. nodorum*. *A. crateriforme* genes and orthologues are coloured sequentially.

rearrangements, which appear to be a hallmark of mesosynteny observed in various fungal taxa[41]. Such high genomic plasticity often led to the high turnover of gene family dynamics or emergence of BGCs capable of producing novel secondary metabolites[42]. In *A. crateriforme*, BGCs were enriched in subtelomeres (12 out of 26 subtelomeres with an observed-to-expected ratio of 4.7). We identified a case of one polyketide cluster located on the end of chromosome 6, which is shared with the plant pathogen *Elsinoe ampelina* and *Parastagonospora nodorum* (Fig. 4d), presumably because of *A. crateriforme* constantly encountering a plant-associated environment.

### Digestion-related genes were co-opted

To dissect how sundews and *A. crateriforme* respond to each other or during insect digestion, we first characterised the baseline transcriptome from both species cultured under minimal nutrient conditions. These baseline data were then compared with two experimental conditions: application of ant powder indicative of insect digestion and fungal inoculation onto sundew leaves replicating coexistence (Extended Data Fig. 8). Around 58.6–63.8% of differentially expressed genes (DEGs) identified between baseline and digestion phase for each species were also differentially expressed in the coexistence phase in the same trend relative to the baseline (Fig. 5a). Gene ontology (GO) analysis revealed that *D. spatulata* upregulated genes in both conditions were enriched in the secondary metabolic process, including response to chemical substances and other organisms (Supplementary Fig. 11 and Supplementary Table 6). This suggests that the majority of plant genes that were involved in defence mechanisms[20,43] have been co-opted in the digestion process but still retained their ancestral functions. An example includes members of plant chitinases (GH18 and GH19) (Supplementary Fig. 12), which have roles in insect digestion but have ancestral functions in phytopathogen defence[44,45]. For nutrient-acquisition-related genes, ammonium transporters, nitrate transporter and nitrate reductase in *D. spatulata* showed increased transcript abundance in both digestion and coexistence phases, and are central to nitrogen uptake and assimilation[17], but were more highly expressed in the coexistence than in the digestion phase (determined by DESeq2 (ref. [46]); |log$_2$(fold change)| > 1 and adjusted *P* < 0.05; Supplementary Fig. 13), suggesting that active nitrogen exchange and use are already taking place within the plant–fungus holobiont.

### Transcriptome dynamic of holobiont digestion in nature

In nature, the digestion of insects takes place in the mucilage of *D. spatulata*, with arthropod remains adhering to the stalk glands, where *A. crateriforme* can be observed growing over the insect surface (Extended Data Fig. 9). To elucidate the mechanisms of carnivorous holobiont digestion in a laboratory setting, we further characterised the holobiont transcriptome following the addition of ant powder (Extended Data Fig. 8) and identified 2,401 and 2,427 DEGs in *A. crateriforme* and *D. spatulata*, respectively. The majority of these genes could be designated to three categories (Fig. 5b)—differentially expressed in single (coexistence or digestion) or similar expression trends in both processes (coexistence and digestion). We also defined a fourth 'additive' category in which the genes were only significantly differentially expressed when stimuli from both the interacting partner and the insect prey nutrient source were present (Fig. 5c). More than half of upregulated DEGs in both species were designated in both or additive categories, suggesting co-evolution and optimization of the plant holobiont transcriptome as a result of constantly encountering each other and insect prey[24]. Within *A. crateriforme*, the highest number of DEGs were categorised as involved in coexistence, suggesting its primary role in species interaction. Only 22.2% of fungal DEGs were differentially expressed in both processes. A BGC on chromosome 6 (Fig. 4d) showed a consistent upregulation in the coexistence and digestion phases (Supplementary Fig. 14) highlighting the need to effectively respond to multiple stimuli in natural environments. Interestingly, GO

term enrichment of condition-specific genes revealed an opposite trend of up- and downregulation of genes involved in the fungal and plant cell cycle, respectively (Supplementary Table 7), suggesting divergent responses in both species when faced with a similar environment.

To investigate the nature of sundew DEGs in these designated categories, we conducted additional leaf transcriptomes inoculated with dead *A. crateriforme* or chitins (Extended Data Fig. 8) and compared them with the baseline transcriptome of sterile leaves. Between 54.4% and 71.0% of the previously mentioned DEGs, identified in either the coexistence phase or both the coexistence and digestion phases, were also differentially expressed in the same manner, respectively, upon exposure to chitin or dead fungal material (Fig. 5d). This suggests that these two categories of DEGs are primarily triggered by the presence of chitin (insect exoskeletons and fungal cell walls), paralleling findings in the Venus flytrap, *Dionaea muscipula*, which showed chitin as a crucial cue for gene expression to set the trap towards a 'posed to capture' mode[47]. Conversely, 26.1–46.2% of the DEGs within the digestion or additive category showed similar expression patterns upon elicitor application, respectively (Fig. 5d). An example was asparagine synthetase, a key enzyme in plant amino acid assimilation[48], which was designated in the upregulated additive category (Supplementary Fig. 15). GO enrichment analysis of the upregulated genes in these categories, not influenced by the presence of elicitors, highlighted genes involved in the auxin-activated signalling pathway and cation transmembrane transport (Supplementary Table 8), suggesting the optimization of the genes repurposed to involve prey digestion[21] or in response to available nutrients towards the end of the digestion process.

### Synergistic expression in fungal peptidases and transporters

To investigate the putative roles in identified DEG gene families, additional transcriptome sequencing was performed in *D. spatulata* towards the end of the digestion process (Extended Data Fig. 8 and Methods). The regulation of peptidases during the different digestion or coexistence phases was determined using weighted gene co-expression network analysis[49], which identified three and nine co-expression modules in *A. crateriforme* and *D. spatulata*, respectively (Supplementary Figs. 16 and 17). The most abundant secreted peptidases in *D. spatulata* belonged to the cysteine (MEROPS[50]: C1) and aspartic families (MEROPS: A1). Droserasin, which has been implicated in digestion[22,30], was contained in two co-expression modules (Supplementary Fig. 17), which differed as genes that were constitutively expressed or were upregulated only during digestion. In contrast, most of the highly expressed peptidases in *A. crateriforme* belonged to a co-expression module harbouring two copies of aspartic peptidase (AC_00151800 and AC_00417900), the entire sedolisin[51] family (14 out of 14 copies; MEROPS: S53) associated with increasing acidity and plant-associated lifestyle[52], and a fungal copy with the aforementioned Neprosin domain (Fig. 5e). These genes showed a synergistic effect in expression in prey digestion during coexistence; for instance, the aspartic peptidases emerged as the dominant and third dominant entities across the entire transcriptome, with their expression levels increasing up to 13-fold compared with either condition (Fig. 5e). The same trend was also observed in potassium, amino acid, oligopeptide and sugar transporters (Supplementary Fig. 18). Taken together, the results indicated *A. crateriforme*'s potential role in facilitating and benefiting from digestion in response to the combined signal of host and nutrient.

### Sundew priming via the JA signalling pathway

The genes in sundews associated with JA signalling pathways were upregulated during both coexistence and digestion phases compared with the baseline (Supplementary Fig. 19), with expressions peaking towards the end of digestion suggesting that the carnivorous plants were primed before prey encounter[53,54]. To examine how sundews respond specifically to either symbiotic microorganisms or prey, quantification of *D. spatulata* leaf phytochromes showed that applications

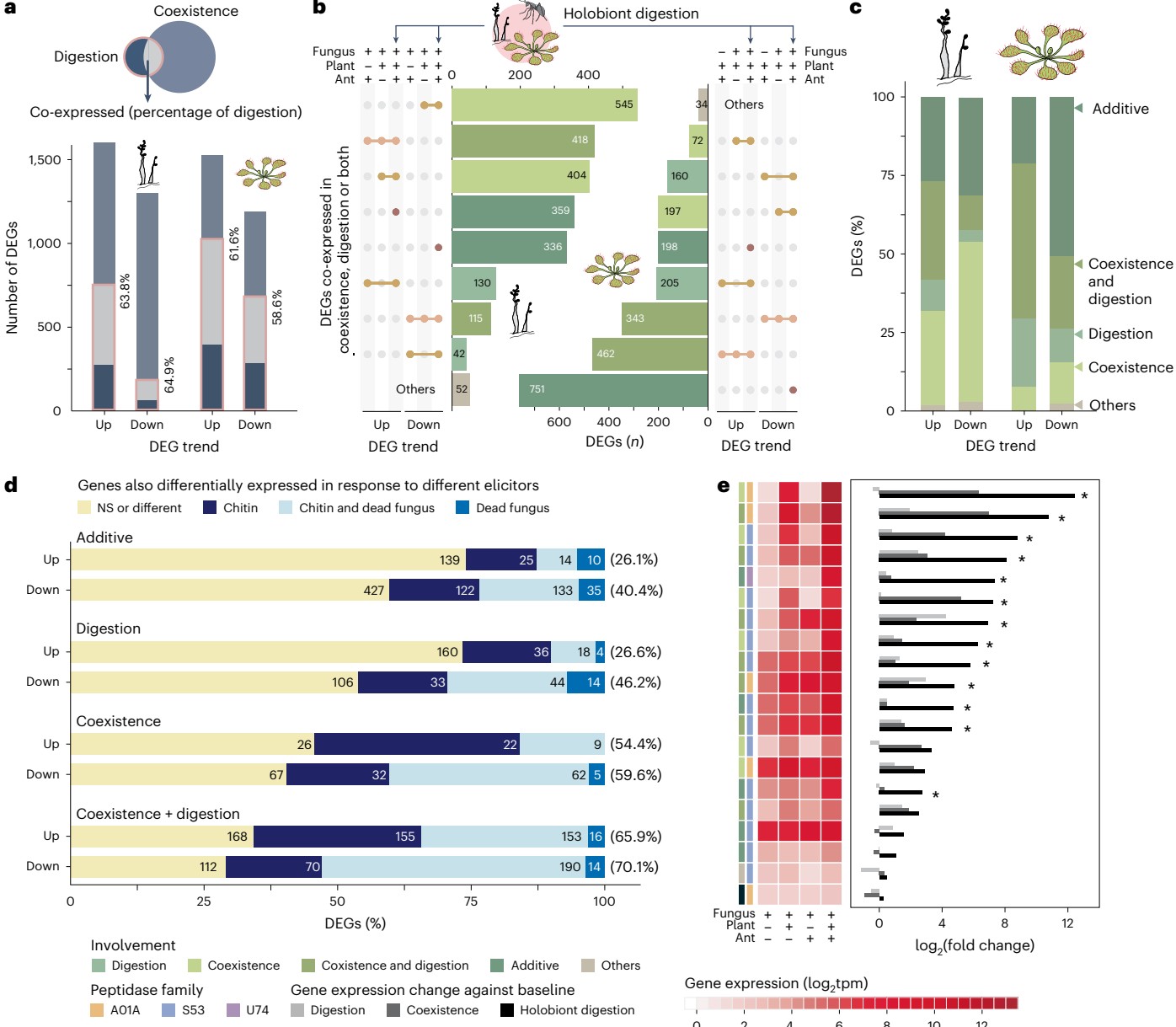

**Fig. 5 | Transcriptome of the *D. spatulata*–*A. crateriforme* holobiont during digestion. a**, Overlap of upregulated (up) or downregulated (down) DEGs during digestion and coexistence for both species compared with baseline controls (determined by DESeq2; |log₂(fold change)| > 1 and adjusted *P* < 0.05). Percentages indicate the proportion of differentially expressed genes in the digestion phase that were also expressed in the same trend in the coexistence phase. **b**, Transcriptome profiling of the plant–fungus holobiont during digestion with the fungus on the right and the plant on the left. DEGs were compared whether the same trends were observed in either the digestion or the coexistence process. **c**, Schematic representation of the role of the DEGs involved in the holobiont digestion. **d**, Schematic representation of designated categories

of DEGs in **c** when compared with transcriptome changes in the plant treated with chitin or dead fungus. Numbers in rectangles denote the number of genes. NS denotes not significant, that is, DEGs that did not exhibit expression changes in response to elicitors. Numbers in brackets denote the proportion of DEGs that were also expressed in the same trend when the plant was treated with different elicitors. **e**, Expression of representative fungal peptidases in a co-expression module (module 2 in Supplementary Fig. 16) showing synergistic effects when both plant and insect prey are present. Asterisks indicate significantly upregulated expression between holobiont digestion and either the digestion or the coexistence phase. tpm, transcript per million.

of ant powder and different fungi significantly increased the amount of jasmonoyl-ʟ-isoleucine (JA-Ile) after 2 h (Supplementary Fig. 20 and Fig. 6) indicative of a potential priming effect. We found that JA levels showed an increasing trend when plants inoculated with *A. crateriforme* were supplemented with ant powder (Supplementary Fig. 21), which correlates with the activation of JA-related priming in plants[54,55]. By contrast, no differences in salicylic acid between the control and treatments were observed (Supplementary Fig. 21). The observed acceleration in trap reopening when plants were inoculated with microorganisms

(Fig. 3d) further underscores the role of *A. crateriforme*-induced JA biosynthesis and signalling in enhancing prey digestion.

## Discussion

This study defines the symbiotic interaction between the carnivorous sundew *D. spatulata* and the acidophilic fungus *A. crateriforme* (Fig. 6), reshaping our view of botanical carnivory since Darwin's foundational work[2]. We show that the insect prey digestion time by sundews was reduced by a quarter with plant priming following successful

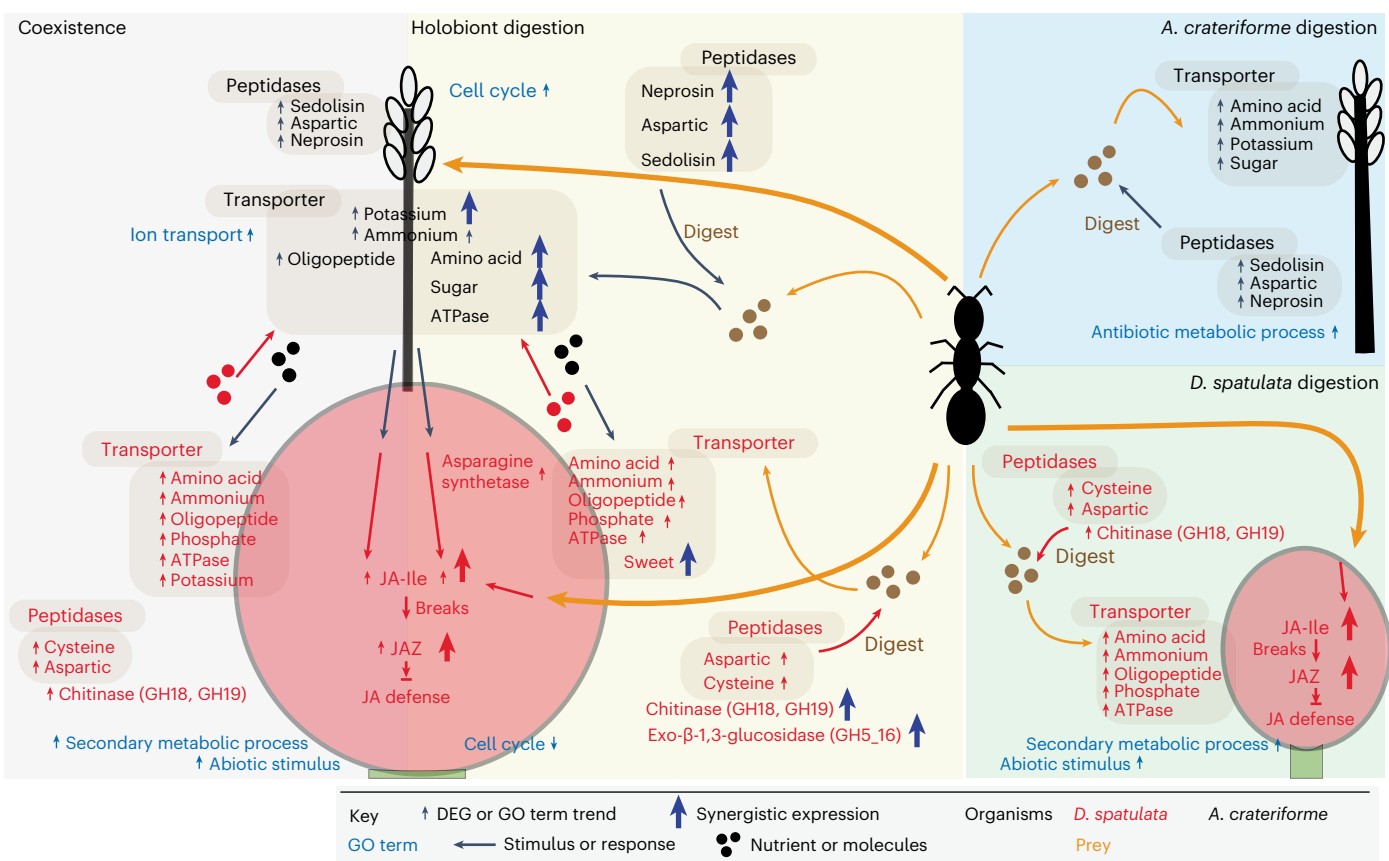

**Fig. 6 | Interactions within the *D. spatulata*–*A. crateriforme* holobiont.** A schematic diagram of a *D. spatulata* stalk gland and an *A. crateriforme* conidiophore with a summary of gene expression changes identified in this study. Genes that are co-expressed in different phases are shown multiple times.

colonization of microorganisms[53,54] and that *A. crateriforme* alone was able to recapitulate and enhance the digestion process including evidence of specific protein degradation. This plant–fungus cooperation is probably rooted in the common adaptive challenges that both carnivorous plants and extremophilic fungi face in extreme environments with limiting nutrients. Detailed dissection of the gene expression in both partners also reveals evolutionary adaptation of multifunctional gene groups to support cooperative prey digestion. This dynamic plant–fungus interaction reveals a multidimensional adaptation that expands significantly the understanding of botanical carnivory.

*A. crateriforme* was previously considered ubiquitous across environments such as soil[56], plant material[57,58], compost[59], air[60] and rock surfaces[61]. It is now recognised as part of an intimate sundew–fungus holobiont with its presence and frequent dominance in several *Drosera* species globally and has probably co-evolved with sundew ancestors. Echoed by observations of selective microbial compositions in adverse environments[62], the chemical restrictions posed by acidic *D. spatulata* mucilage could potentially constrain the spectrum of acid-tolerant microorganisms possible for digestive collaboration. As *A. crateriforme* was able to survive in this extreme environment, this may be a first step in establishing a long-term, stable relationship. The dominance of this fungus in the *Drosera* microbiome, which was not observed in most of the other carnivorous plants, may also be correlated with relative osmotic stress caused by air exposure of the stalked glands, a condition not observed in the traps of other carnivorous plants, which contained a liquid medium to a greater extent[12,63,64].

We propose that, following colonization of leaf glands, *A. crateriforme* underwent a series of genomic changes in adapting to the symbiotic lifestyle[38,39], providing functional benefits by reducing digestion time and allowing more rapid reopening of traps, thus increasing prey capture rates over time. Significant degradation of BSA in the presence of *A. crateriforme* and numerous genes including peptidases showing synergistic expression in *A. crateriforme* indicates that the fungus facilitates the digestion processes in the shared environment[24]. For example, the entire upregulated fungal sedolisin family in *A. crateriforme* has been associated with higher acidity and a plant-associated lifestyle in other fungi[52]. The combined peptidases from both species can degrade large proteins and peptides on acidic mucilage to generate nutrients in decomposing insect prey. This may ultimately result in more digested nutrients to both the fungus and plant host, which was evident from upregulation of sundew's gene families involved in nitrogen assimilation such as the asparagine synthetases, and transporters that were upregulated only during prey digestion with *A. crateriforme* present. Together, these results also imply that the plant–fungal coexistence is cooperative and may be mutualistic, as prey capture rates are positively associated with plant fitness[65] and may be especially relevant in *Drosera*, which are carnivorous plants considered sit-and-wait predators. Considering that the nutrient acquisition efficiency of carnivorous plants from prey is usually low (29–42% from available nitrogen[66]), it is likely that there is sufficient N available to *A. crateriforme* and other microorganisms present.

It is generally accepted that plant carnivory genes are evolved from defence mechanisms. Our transcriptomic comparisons of *A. crateriforme* in single- or dual-species partnerships during the digestion phase, however, enabled us to delineate the relative contribution of fungus versus plant to digestion as well as to identify key genes in the process. The high extent of genes maintained similar expression in both the digestion and coexistence phases in *D. spatulata* suggesting that the sundew uses a shared gene pool for these processes. In the presence of abiotic and biotic stimuli, many plants will enter a priming

phase and are known to boost defensive capacity against pathogens[53]. In the case of sundews, colonization of leaves by *A. crateriforme* enables the carnivorous plants to induce transcription changes before the prey is captured. This is akin to the 'posed to capture' phase shown in the Venus flytrap *D. muscipula*[47], or successful mycorrhization[54]. As almost half of the genes involved in digestion were already modulated in the priming phase, actual digestion of prey required transcription of fewer genes to be induced resulting in an overall faster digestion process.

In summary, we have isolated a keystone fungus and provide direct evidence that *A. crateriforme* is integral to prey digestion in carnivorous sundews (Fig. 6), having been restricted so far to observing microbiota changes in carnivorous plants in nature or via the addition of prey (Extended Data Table 1). This study provides evidence for a specific fungal–plant collaboration that has apparently undergone selection to increase the overall fitness of the holobiont[24]. Just as microbial communities in carnivorous pitcher plants provide systems for addressing microbial ecology, and food webs and evolution[67], this *Drosera–Acrodontium* partnership provides an amenable laboratory system for exploring plant–microbial partnerships, as both can be grown separately and together in the laboratory. This study supports the hypothesis that plant–microbial interactions facilitating the insect prey digestion are functioning in different carnivorous plants, supporting prey breakdown processes.

## Methods

### Collection of sundew mucilage and surrounding plants

We collected *D. spatulata* mucilage samples from three collection sites (Shumei, Shuangxi and Yangmingshan) located in northern Taiwan (Supplementary Fig. 2). A typical habitat of *D. spatulata* is shown in Supplementary Fig. 1, which usually harbours hundreds of sundew plants on a cliff surface. A sample consisting of pooled mucilage was collected from all leaves (usually 15–20 leaves per plant) of 30 *D. spatulata* in the same site by using filter papers of size 1 cm × 1.5 cm. Multiple samples within the same site were separated by at least 5 m. The leaf surfaces of plants adjacent to sundews were also sampled. Targeted plants were usually mosses, grasses and some vascular plants. Leaves were wiped and considered as environmental samples for that site. To determine the temporal dynamics of fungi in mucilage, we repeated the sampling in Shumei and Shuangxi sundew mucilage monthly from June 2018 to April 2019.

To understand the spatial extent of plant–fungus coexistence, we repeated the mucilage sampling from 17 additional sites encompassing a 44 km radius in northern Taiwan (Supplementary Table 1) for 52 additional samples.

### Sampling and surface sterilization of US and UK *Drosera* samples

Sites were surveyed for *Drosera* distribution, and plants were collected from across the entire range; at Cedarburg, plants were sampled from a boardwalk loop on the bog. The plants selected were free from visible infection or damage, and leaves were carefully picked and placed individually in Ziplock bags and then stored in a cool box for transportation and at 4 °C in the laboratory.

Within 24 h of collection, the plants were surface sterilised to remove any fungi contaminating the plant surface. Samples were handled in biological safety cabinets to reduce contamination. Any prey or debris was removed from the leaves, which were surface sterilised in a three-step approach[68]: first submerged in 95% ethanol for 30 s, then immediately transferred to 10% bleach solution for 180 s and finally submerged in 70% ethanol for 60 s. Leaves were then rinsed in ultrapure water (18.2 MΩ cm) and left to air-dry at room temperature, under a biological safety cabinet. Dried leaves were stored in individual Ziplock bags with silica gel beads to preserve internal fungal DNA for extraction. Over 2 weeks, the silica was replaced to maintain dry conditions.

### Genomic DNA extraction and metabarcoding of Taiwanese environmental samples

The total genomic DNA of Taiwanese samples was extracted from filter papers using modified cetyltrimethylammonium bromide (CTAB) DNA extraction protocol. For cell lysis, 5 ml CTAB buffer (0.1 M Tris, 0.7 M NaCl, 10 mM EDTA, 1% CTAB, 1% beta-mercaptoethanol) was added to a 15 ml tube containing the sample. After incubation at 65 °C for 30 min, an equal volume of chloroform was added. The mixture was centrifuged at 10,000 g for 10 min, and the supernatant was mixed with an equal volume of isopropanol. After centrifugation at 10,000 g for 30 min at 4 °C, the supernatant was discarded and the pellet was washed twice with 70% and 90% ethanol. DNA was eluted with 50 μl elution buffer (Qiagen).

ITS and 16S rRNA amplicons were generated using barcode primer pairs ITS3ngs(mix)/ITS4 (ref. 69) and V3/V4 (ref. 70), respectively. Amplicon levels were standardized using the SequalPrep Normalization Plate 96 Kit (Invitrogen, catalogue number A10510-01). The concentration of pooled and standardised amplicons was performed using Agencourt AMPure XP beads (Beckman Coulter, catalogue number A63881). All amplicon libraries were sequenced with Illumina MiSeq PE300 using 2 × 300 bp paired-end chemistry performed by the NGS High Throughput Genomics Core at the Biodiversity Research Center, Academia Sinica, Taiwan.

### Genomic DNA extraction and metabarcoding of UK and US *Drosera* samples

Dried plant tissue approximately 0.02 g was transferred to individual 1.5 ml tubes and homogenised using a single stainless-steel bead in each tube and shaking for 2 min in a bead mill tissue lyser. DNA was extracted from homogenised tissue using a NucleoSpin Plant II kit (Macherey-Nagel), following the manufacturer's protocol PL2/PL3 pathway, with a cell lysis time of 1 h in a rotating oven and 50 μl elution volume.

PCR amplifications were carried out with primers covering the ITS1 region, the forward primer ITS5 (GGAAGTAAAAGTCGTAACAAGG)[71] and the reverse 5.8S_Fungi (CAAGAGATCCGTTGTTGAAAGTK)[72] in 20 μl reactions with the following reagents: 10 μl Qmix, 2 μl forward primer, 2 μl reverse primer and 4 μl double-distilled $H_2O$ (ddH_2O), with 2 μl DNA. A touchdown PCR cycle was used to amplify fungal DNA and reduce primer dimers. The thermocycler programme was as follows: an initial activation step of 600 s at 95 °C, followed by 10 cycles of 20 s at 95 °C, 60 s at 61–55.6 °C and 60 s at 72 °C. After each cycle, the annealing temperature was decreased by 0.6 °C, starting at 61 °C and finishing at 55.6 °C. After the touchdown step were 30 cycles of 30 s at 95 °C, 30 s at 55 °C and 60 s at 72 °C. The final step was then 420 s at 72 °C.

Amplicons were pooled into libraries, each containing amplicons from eight samples; libraries were then cleaned using ProNex beads (Promega) and BluePippin (SageScience) following manufacturer protocols to select fragments of 200–500 bp. All libraries were then pooled into a single sequencing library; volumes were used such that each sample was present at 4 nM. Amplicons were sequenced on a paired-end (2 × 250 bp) Illumina Miseq platform at the Natural Environment Research Council (NERC) Environmental Omics Facility (Liverpool, England).

### Amplicon analysis

Samples were demultiplexed using Sabre, allowing one nucleotide mismatch (v1.0; https://github.com/najoshi/sabre) with their respective barcodes. Adaptor and primer sequences were trimmed using USEARCH[67] (v11.0.667). Sequence reads were processed according to the UPARSE[73] pipeline. Before data analysis, the accuracy of the UPARSE pipeline was assessed using positive controls containing known species. Forward and reverse reads were merged and filtered using USEARCH. OTUs were clustered at 97% sequence identity with a minimum of 8 reads per cluster to denoise the data and remove singletons.

The OTU table was generated using the 'otutab' option and analysed with the 'phyloseq'[74] package (v1.46.0). The OTU sequence taxonomy was classified using 'constax'[75] (v2.0.19) against the SILVA database (v138.1) and the UNITE[76] database (v9). Taxonomy[74,77–81] levels of ITS and 16S OTU were updated using the 'rgbif'[3] package (v3.7.7). OTUs containing less than the numbers of reads present in the negative controls of different sequencing runs were removed from subsequent analyses. There were four datasets that were analysed independently: (1) the initial survey of microbial communities of *D. spatulata* mucilage in northern Taiwan, (2) the fungal community in mucilage across two sites at a temporal scale, (3) the broader survey of fungal communities of *D. spatulata* mucilage in northern Taiwan and (4) the fungal community in multiple *Drosera* species across the United Kingdom and the United States. For each analysis, samples were rarefied according to the sample with the fewest sequences if it exceeds 10,000 reads or capped at 10,000 reads. Fourteen UK samples and one Taiwanese sample were removed owing to an insufficient number of sequences.

### Preparation of *D. spatulata* stalks for SEM

*D. spatulata* leaf tissues were fixed with P4G5 solution (4% paraformaldehyde and 2.5% glutaraldehyde in 0.1 M phosphate buffer) at 4 °C for 1 h. After three washes with 0.1 M phosphate buffer, secondary fixation was performed in a 1% solution of osmium tetroxide for 2 h at room temperature. The fixed samples were passed through a series of dehydration steps in 30%, 50%, 70%, 80%, 90%, 95%, 100%, 100% and 100% alcohol before drying in a critical-point dryer (Hitachi, model HCP-2). The dried samples were then coated with a layer of gold using a sputter coater (Cressington, model 108).

### Morphology observation of the *D. spatulata* stalk gland

The stalk glands of *D. spatulata* were photographed using a microscope (Olympus CX31) and a digital camera (Nikon D7000) following staining with cotton blue reagent[82]. SEM of the *D. spatulata* stalks were prepared. *Drosera* leaves grown under different conditions (laboratory condition, inoculated with *A. crateriforme* and from the wild) were observed with a JSM-7401F scanning electron microscope (JEOL) at the Institute of Plant and Microbial Biology, Academia Sinica, Taiwan.

### Isolation and identification of fungal species from sundew mucilage

Mucilage-soaked filter papers were placed in potato dextrose broth containing chloramphenicol (50 µg ml$^{-1}$) and incubated for at 30 °C for 24 h. The broth culture was then serially diluted ($10^{-2}$, $10^{-3}$, $10^{-4}$), and 200 µl of the diluted culture was transferred to PDA. The plate was spread on sterilised glass until the medium was dry and incubated at 30 °C for 24 h. Single colonies were transferred onto fresh PDA using sterilised toothpicks and examined for morphology using light microscopy. Genomic DNA was extracted from the fungal pure cultures corresponding to the morphological description of *Acrodontium*, followed by amplification and sequencing of the fungal ITS region as previously described[79,80]. To construct a phylogenetic tree, ITS sequences of the sequenced 18 isolates, 15 *Acrodontium*, *Teratosphaeria biformis* and *Aureobasidium pullulans* ITS sequences from the National Center for Biotechnology Information were first aligned using MAFFT[81] (v.7.4) and trimmed using trimAl[83] (v.1.2). A maximum likelihood phylogenetic tree with 100 bootstraps was produced from the alignment using IQtree[84] (v.1.6.1). All 18 isolates were classified as *A. crateriforme* based on morphological description and grouped with the *A. crateriforme* ITS sequence.

### Sterile laboratory culture of *D. spatulata*

*D. spatulata* seeds were collected from the sampling sites and sterilised by washing with 70% ethanol for 10 s and 3% (w/v) calcium hypochlorite (CaCl$_2$O$_2$) for 30 s and finally rinsed 3 times with ddH$_2$O. Surface-sterilised seeds were pregerminated on 0.5% water agar and

incubated in the dark at 20 °C for 6–8 weeks in sterile conditions. Shoots were then transferred to 1/2 MS agar, with the pH adjusted to 5.7. Shoots were grown under white fluorescent light at 20 °C with a 16 h:8 h, light:dark photoperiod for 90 days as recommended[85].

### Growth profiling of *A. crateriforme* and *D. spatulata*

*A. crateriforme* was inoculated in a PDA medium agar plate at different temperatures (20 °C, 25 °C, 30 °C) and pH values (pH 3–9) for 2 weeks. We added ant powder to 1/2 MS medium agar plates and adjusted the concentration to 1 g (ant powder) per l (1/2 MS medium). *A. crateriforme* was then inoculated in this medium with or without ant powder and cultured for 2 weeks at 25 °C without light. The fungal growth area was photographed every 2 days on plates using a Nikon digital camera and measured using ImageJ[86].

Photographs of *D. spatulata* in different treatments including control plants were taken daily, and the area of leaves showing symptoms of infection was quantified every 2 days using ImageJ[86]. Whole *D. spatulata* plants were dried with tissue paper and weighted on the 1st and 30th day of the experiment.

### Inoculation of different microbiota and growth of *D. spatulata* in the lab

Filter papers were used to collect mucilage from leaves of individual *D. spatulata* plants or used to wipe the leaf surfaces of surrounding plants. A single filter paper thus consisted of the microbiota of pooled mucilage of a single sundew or entire leaf surfaces of a single adjacent plant. For inoculation of a pure fungal strain, conidia were removed from 14 day fungal colonies grown on PDA medium by washing with sterile distilled water and the suspension was diluted to ~10$^6$ spores per ml.

Sterile *D. spatulata* plants were grown from seeds and were transferred from the 1/2 MS agar (2.22 g MS basal medium powder, 0.5 g MES sodium salt, 30 g sucrose and 3 g Phytagel per l) to vermiculite with ddH$_2$O and incubated for 4 weeks in the greenhouse with a 16:8 light:dark photoperiod at 25 °C before inoculation. Sampled filter papers were picked with a sterile pair of tweezers and dipped into the mucilage of all leaves on a single sundew plant on the day of collection. Each leaf was dipped 20 times. In addition, the prepared fungal spore solution was used to inoculate every leaf of a sundew plant with a pipette. The inoculated plants were grown in an incubator with a 16:8 light:dark photoperiod at 25 °C for 30 days.

### Preparation of different substrates for feeding experiment

Ant individuals (*P. dives*) were collected from the National Taiwan University campus. The collected ants were washed using distilled water (ddH$_2$O) and subsequently dried for 2 days. Subsequently, the ants were frozen in liquid nitrogen and then crushed to powder form. Wood (*Populus tristis*) was provided by Prof. Ying-Chung Jimmy Lin's laboratory at National Taiwan University. We used dissecting scissors to cut the wood into 0.4 cm × 0.4 cm × 0.1 cm pieces. Shrimp shells (*Litopenaeus vannamei*) were collected from commercial shrimp (Kirkland Signature Frozen Raw Tail-On Shrimp, number 7777000). The shrimp tissues were removed, and the shells were washed with sterile distilled water for 20 min. Subsequently, shrimp shells were cut into pieces measuring 0.4 cm × 0.3 cm using dissecting scissors (D8NH-76000). Prepared substrates were autoclaved at 121 °C for 40 min and dried at 60 °C in a forced-air-flow oven (DV-092) for 2 days. Then, 1 g of ant powder was mixed with 1 l of ddH$_2$O to use in a subsequent experiment.

### Feeding experiment of *D. spatulata* containing different microbiota

A total of 10 µl of ant solution containing 100 µg of ant powder (*P. dives*) was added to a single leaf of *D. spatulata* with different microbiota inoculated with a pipette. Then, 2.5 mg wood powder (*P. tristis*)

and 1 mg shrimp shell powder (*Pleoticus muelleri*) were weighed and added using tweezers. The sundew leaf was gently poked with tweezers until it closed. Each replicate was a single leaf from a sundew and each comparison consisted of five leaves from five independent sundew plants. The time from addition of substrates until the tentacles fully closed then reopened was recorded. This procedure was repeated with additional application of Protease Inhibitor Cocktail (number P9599, Sigma) using modified substrates containing 0.99 µl of ant medium mixed with 0.01 µl Protease Inhibitor Cocktail.

## BSA degradation using mucilage from sundews with different microbiota

To prepare BSA labelled with Sulfo-NHS-Biotin, we first prepared 1 mg of BSA in 1 ml of PBS. Then, we added 500 µl ultrapure water to 2.2 mg Sulfo-NHS-Biotin, then added 30.3 µl of a 10 mM biotin reagent solution to 1 ml BSA solution. The reaction was incubated at room temperature for 30 min and precipitated by sixfold volume of acetone at −20 °C overnight. The reaction was then centrifuged at 12,000 $g$ for 10 min at 4 °C. The supernatant was carefully removed, and the protein pellet was diluted in PBS to a concentration of 1 mg ml$^{-1}$.

To compare the effect of protein digestion in mucilage from *D. spatulata* with different microbiota, pooled mucilage from 30 plants in each treatment (sterile *D. spatulata* plants, *D. spatulata* inoculated with *A. crateriforme*, wild *D. spatulata* mucilage and co-occurring plant microbiota) was collected before the experiment, mixed with 0.01 mg BSA labelled with Sulfo-NHS-Biotin and incubated at 25 °C for 16 h or 24 h. The protein remaining after the digestion during incubation was imaged using a western blot with horseradish peroxidase-conjugated streptavidin antibodies (dilution rate, 1/5,000×), and the amount of protein remaining was quantified using ImageJ[86] and used to calculate the digestion.

## Western blot

We mixed 10 µl protein sample with 10 µl of 2× loading dye in a 0.2 µl PCR tube. The samples were then incubated for 10 min at 99 °C in the PCR machine. After incubation, the protein samples were loaded onto the sodium dodecyl sulfate polyacrylamide gel electrophoresis (SDS-PAGE). Electrophoresis used 90 V for 20 min in the stacking gel and then used 120 V for 120 min in the running gel. After electrophoresis, we transferred the SDS-PAGE into a 1× transfer buffer for 10 min. Subsequently, we put soaked filter papers, membrane, and SDS-PAGE together. The machine parameters for transferring were set to 0.8 mA cm$^{-2}$ for 1 h. We washed the membrane with 1× phosphate buffered saline with Tween-20 (PBST) for 5 min twice. Then, the membrane was soaked in blocking buffer at room temperature for 1 h and washed twice with 1× PBST for 5 min. We added the HRP-conjugated streptavidin and subjected the membrane to shaking for 30 min at room temperature. The membrane was washed with 1× PBST for 8 min at room temperature three times. The luminol substrate and peroxidase in a 1:1 ratio (750 µl + 750 µl) were added, and the membrane was placed into the detecting chamber for signal detection using the iBright CL750 Imaging System (Thermo Fisher Scientific, A44116). An initial western blot was produced to test the condition in Supplementary Fig. 5, and all the subsequent blotting images are shown in Supplementary Figs. 22–24.

## RNA extraction

In addition to the plants already inoculated with *A. crateriforme*, we also inoculated *D. spatulata* with dead *A. crateriforme* spores and chitin. We diluted *A. crateriforme* conidia into 10$^6$ spores per ml and prepared 1 g l$^{-1}$ chitin solution with ddH$_2$O. The prepared solutions (death fungus, chitin) were autoclaved at 121 °C for 40 min and used to inoculate every leaf of a sundew plant with a pipette. The inoculated plants were grown in an incubator with a 16 h light–8 h dark photoperiod at 25 °C for 30 days. A schematic diagram of different plant treatments is shown in Extended Data Fig. 8.

We pooled 80 leaves as one replicate and each treatment had five replicates. At each harvest, leaf tissue was cut from the plant and washed immediately in deionised water to remove prey residue and, within 30 s, the leaf was flash-frozen in liquid nitrogen (−196 °C). Total RNA was extracted using the modified CTAB method. Then, 2 ml CTAB buffer (0.1 M Tris, 2 M NaCl, 25 mM EDTA, 2% CTAB, 1% PVP-40, 2% beta-mercaptoethanol) was added into a 2 ml tube containing a leaf sample. Samples were frozen in liquid nitrogen and ground with a Precellys 24 tissue homogenizer. After incubation at 65 °C for 20 min, an equal volume of chloroform–isomylalcohol (24:1) was added. The mixture was centrifuged at 12,000 rpm for 10 min twice. Supernatants were mixed with 1/3 volume of LiCl and kept at 4 °C overnight to precipitate RNA. After centrifugation at 10,000 rpm for 30 min at 4 °C, the supernatant was discarded and the pellet was washed twice with 70% ethanol. RNA was resuspended in 30 µl diethyl pyrocarbonate (DEPC) water. RNA samples were sequenced with the Illumina Novaseq 6000 sequencer.

## Genome sequencing, assembly and annotation of *A. crateriforme*

Genomic DNA was subjected to Oxford Nanopore library preparation according to the manufacturer's instructions (SQK-LSK109) and sequenced on a GridION instrument. Basecalling was done using Guppy (version 3.0.3). For Illumina sequencing, genomic DNA was used for NEB Next Ultra library preparation and 150 bp paired-end reads were generated on a Novaseq 6000 sequencer. The nanopore reads were first corrected from the initial assembly of the Canu[87] assembler (version 1.9), which were then assembled using the Flye assembler (version 2.5). The initial assembly was polished by Racon[88] (four iterations; version 1.4.11), followed by Medaka (version 0.11.0; https://github.com/nanoporetech/medaka) using nanopore reads and Pilon[89] (version 1.24) with Illumina reads. The mitochondrial genome was assembled separately using NOVOPlasty[90] (version NOVOPlasty2.7.0.pl).

## Gene prediction of *A. crateriforme*

The transcriptome reads were mapped to the *A. crateriforme* genome assembly using STAR[91] (version 2.7.7a) and assembled using Trinity[92] (version 2.13.2; guided approach), Stringtie[93] (version 2.1.7) and Cufflinks[94] (version 2.2.1). Transcripts generated by Trinity were mapped to the assembly using Minimap2 (ref. 95) (version 2.1, options: -ax splice), and splice junctions were quantified using Portcullis[96] (version 1.2.3). The gene predictor Augustus (version 3.4.0) and gmhmm[97] were trained using BRAKER2 (ref. 98) (version 2.1.6) and SNAP[99] (version 2006-07-28) with proteomes and RNA-seq mappings as evidence hints to generate an initial set of annotations. The assembled transcripts selected by MIKADO[100] (version 2.3.3), proteome downloaded from Uniprot Fungi (version October 2019) as homology and BRAKER2 annotations were combined as evidence hints for input into the MAKER2 (version 3.01.03) annotation pipeline[33] to produce a final annotation for each species. Repetitive elements were identified based on protocol from a previous study[101] and masked using Repeatmasker[102] (version 4.1.2).

Functional domains within the proteomes were identified using pfam_scan[103] (version 1.6) against the downloaded Pfam database[103] (version 36). Diamond[104] (version 2.1.6) was used to identify transporters, blasting proteomes against TransportDB[105] (version 2.0). Proteomes were functionally annotated to identify carbohydrate-active enzymes (CAZy) and peptidases using dbCAN[106] (version 2.0.11) and MEROPS[50] (version 12.4), respectively. Annotations of BGC regions and GO terms were performed using antiSMASH[107] (fungi version 7.0.1) and eggNOG[108] (version 2.1.12), respectively.

Analysis and visualizations were conducted under R[109] environment (version 4.3.1). Various packages were used during the analysis: topGO[110] (version 2.52.0) for GO enrichment analysis, pheatmap[111] (version 1.0.12) and ggplot2 (ref. 112) (version 3.4.4) for gene expression visualization.

## Comparative genomics and phylogenomics

A total of 32 genomes from representative fungi and plants were downloaded from the JGI and NCBI databases (Supplementary Table 5). For each gene, only the longest isoforms were selected for subsequent analysis. Orthogroups (OGs) were identified using Orthofinder[113] (version 2.5.5). For each orthogroup, an alignment of the amino acid sequences derived from each gene sequence was produced using MAFFT[81] (version 7.741). A maximum likelihood orthogroup tree was made from the alignment using IQtree[84] (version 2.2.2.6). A species phylogeny was constructed from all orthogroup trees with ASTRAL-III (ref. [114]) (version 5.7.1). OG gains and losses at each node of the species phylogeny were inferred using DOLLOP[115] (version 3.69.650).

## Transcriptome analysis

Total RNA was extracted from 0.1 g leaf samples of *D. spatulata* with different inoculations and used for transcriptome sequencing. RNA-seq raw reads were trimmed using fastp[116] (version 0.23.2) to remove the adaptor and low-quality sequences. The trimmed reads were mapped to the corresponding genome using STAR[91] (version 2.7.10b) and assigned to gene count using featureCounts[117] (version 2.0.3,). Notably, the reads from experiments containing two species were mapped to both *A. crateriforme* and *D. spatulata* genomes[21]. Sequences mapped to both genomes and having low mapping qualities (averaging less than 0.01% per sample) were excluded from further analyses (Supplementary Table 11). The DEGs of different conditions compared with the control were inferred using DESeq2 (ref. [46]) (version 1.38.3; adjusted $P < 0.05$ and $|\log_2(\text{fold change})| > 1$). The gene ontology enrichment of comparisons was identified using topGO (version 2.50.0). We also performed weighted gene co-expression network analysis to further categorise the expression patterns of peptidases respectively in *A. crateriforme* and *D. spatulata*. Before the construction of the network, the 30% lowest-expressed genes in each transcriptome using the sum of samples were excluded. The descriptions and annotations of every DEG in *A. crateriforme* and *D. spatulata* are available in Supplementary Tables 9 and 10.

## Phytohormone analysis

Different combinations of BSA (A7906, Sigma), chitin (C7170, Sigma), ant powder and fungus (1 g l$^{-1}$ of BSA, chitin and ants; 10$^6$ spores per ml of *A. crateriforme*; and 10$^5$ spores per ml of *P. herbarum*) were applied to *D. spatulata* leaves, and the leaves were incubated for 2 h. The leaves were then cut and washed in deionised water to remove residue before flash-freezing for 30 s in liquid nitrogen. The prepared samples were then ready for metabolite extraction.

Metabolites were extracted in 1 ml CHCl$_3$:MeOH (2:1) with dihydrojasmonic acid (H$_2$JA; 7.5 ng for 0.3 g leaf tissue) added as an internal standard. The mixture was mixed at 4 °C for 30 min with a tube revolver (Thermo Fisher Scientific, number 88881002), and the samples were centrifuged twice at 10,000 *g* for 10 min. The supernatant samples were dried using Centrifugal Evaporator (EYELA, CVE-3110) and stored at −80 °C. Each sample was reconstituted in 100 µl of 50% aqueous methanol and analysed in a Vanquish UHPLC system coupled with a Dual-Pressure Ion Trap Mass Spectrometer (Velos Pro, Thermo Fisher Scientific) or an ACQUITY Premier UPLC coupled with a triple quadrupole mass spectrometer (Xevo TQ Absolute, Waters). JA-Ile, jasmonic acid (JA), salicylic acid (SA) and their standard H$_2$JA and d$_6$SA were separated using an HSS T3 column (Waters ACQUITY HSS T3; 100 Å, 1.8 µm, 100 × 2.1 mm) at 40 °C using mobile buffer consisting of 2% ACN and 0.1% FA (buffer A) with an eluting buffer of 100% ACN and 0.1% FA (buffer B) with an 11 min gradient of 0.5–30% buffer B at 0–6 min, 30–50% B at 6–7 min, 50–99.5% B at 7–7.5 min and 99.5–0.1% B at 9.5–10 min and then equilibrated by 0.1% B at 10–11 min. The selected m/z 322.20 to 130.09 for JA-Ile, 209.12 to 59.01 for JA, 137.02 to 93.03 for SA, 141.05 to 97.06 for d$_6$SA and 211.13 to 59.01 for H$_2$JA (ref. [118]).

## Reporting summary

Further information on research design is available in the Nature Portfolio Reporting Summary linked to this article.

## Data availability

All sequences generated from this study were deposited on NCBI under BioProject PRJNA1034788 and PRJNA1095839. The genome and annotation of *A. crateriforme* are available in NCBI under accession GCA_033807595.1. Accession numbers of individual samples can be found in Supplementary Tables 1 and 11. Source data are provided with this paper.

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

## Acknowledgements

We thank S.-Y. Ting, S. F. Sheng and C.-L. Chung for initial discussions and encouragements. We thank K.-H. Chen for reading and commenting on the paper. We thank H.-H. Lee for producing an initial version of the *A. crateriforme* assembly. We thank M. Hasebe for advice on experiments on *D. spatulata*. I.J.T. and P.-F.S. thank B.J.P.S. for reaching out during the preprint stage, which resulted in collaboration. Support for this work was provided by the National Science and Technology Council (112-2628-B-001-005-) and Academia Sinica (AS-IA-113-L04) of Taiwan to I.J.T. P.-F.S. is supported by the doctorate fellowship of the Taiwan International Graduate Program (TIGP), Academia Sinica of Taiwan. E.B.Y. is supported by a United States National Science Foundation award (2025337). B.J.P.S. is supported by a Central England NERC Training Alliance (CENTA) PhD studentship (NE/S007350/1) and the UK NERC Environmental Omics Facility (NEOF; NEOF1465).

## Author contributions

I.J.T. conceived the study. P.-F.S. carried out the sampling and the experiments with help from H.-W.C., Y.-F.L., H.-M.K., Y.-C.J.L. and Y.-L.C. B.J.P.S. and E.B.Y. carried out sampling in the United Kingdom and United States, and B.J.P.S. completed the initial UK and USA amplicon analysis with help from J.M. and E.B.Y. P.-F.S. and C.-P.L. performed and analysed the amplicons with guidance from Y.-F.L. and D.Z.H. M.-Y.J.L. and B.J.P.S. sequenced the amplicons. P.-F.S. and R.K. identified the *A. crateriforme* strains. H.-M.K. carried out Oxford nanopore (ONT) sequencing. I.J.T. carried out *A. crateriforme* genome assembly and annotation. P.-F.S., M.R.L., Y.-C.L. and I.J.T. carried out comparative genomic and transcriptomic analyses. P.-F.S., I.-F.W. and Y.-L.C. quantified the phytohormones. P.-F.S. and I.J.T. wrote the paper with input from all authors. All authors read and approved the final paper.

## Competing interests

The authors declare no competing interests.

## Additional information

**Extended data** is available for this paper at https://doi.org/10.1038/s41564-024-01766-y.

**Correspondence and requests for materials** should be addressed to Isheng Jason Tsai.

**Reviewer Recognition** *Nature Microbiology* thanks Rainer Hedrich, Jason Stajich and the other, anonymous, reviewer(s) for their contribution to the peer review of this work.

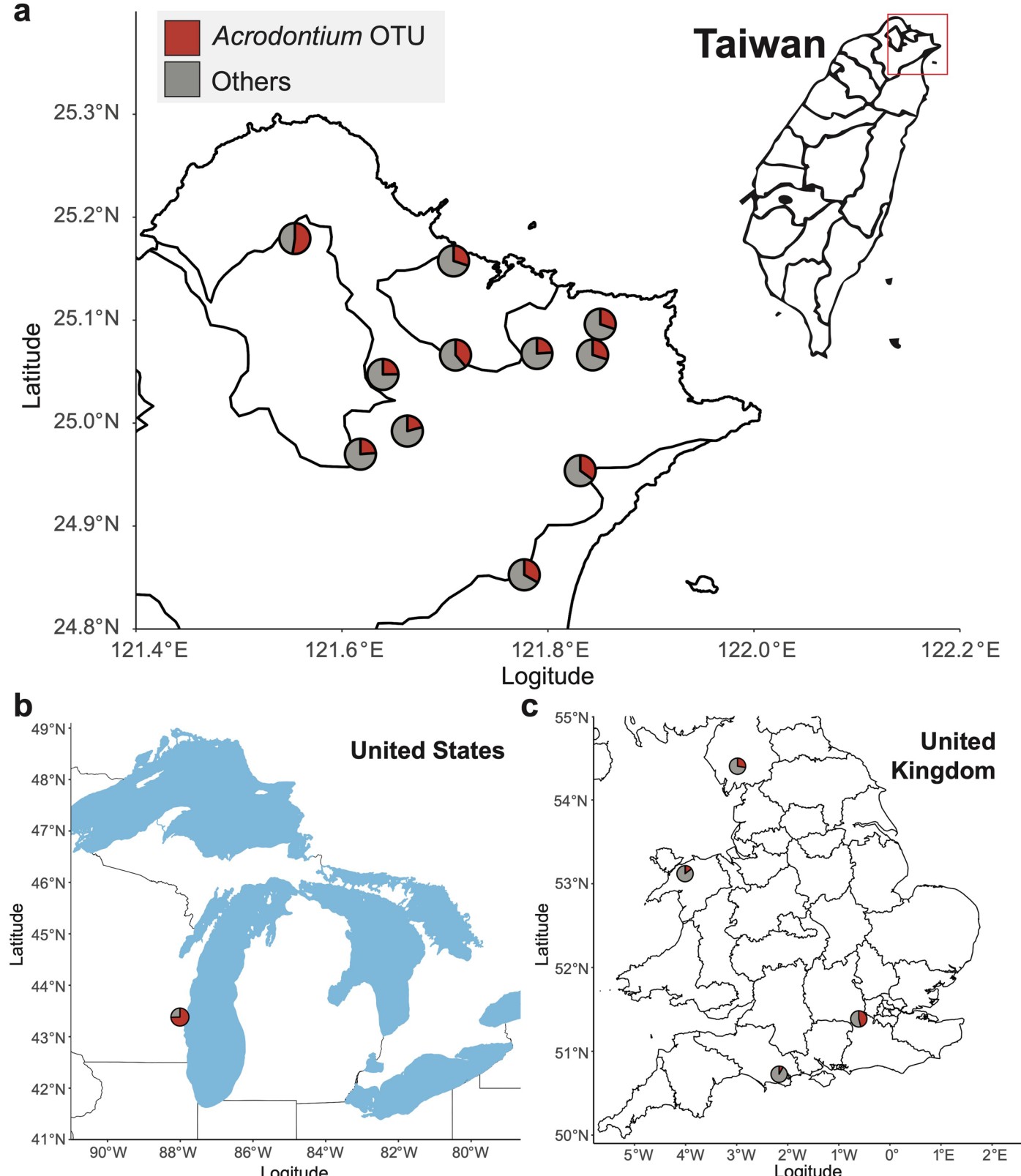

**Extended Data Fig. 1 | Abundance and spatial distribution of *Acrodontium crateriforme* in a. Taiwan, b. USA and c. UK.** Pie chart denote relative abundance of fungal OTUs.

a

b

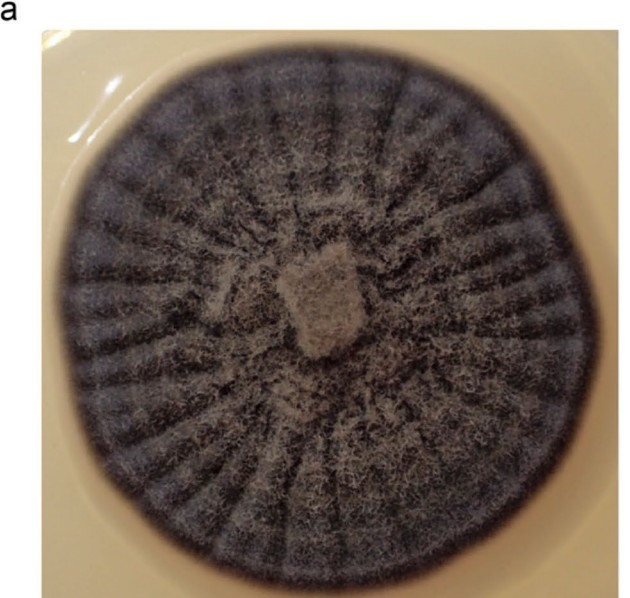
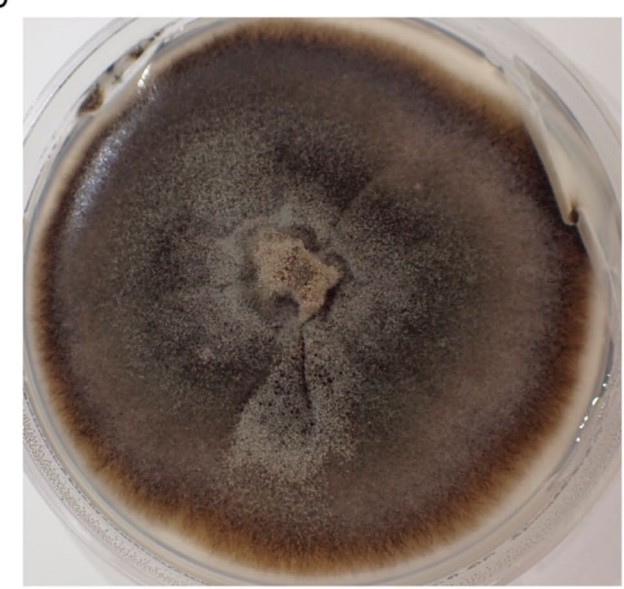

c

d

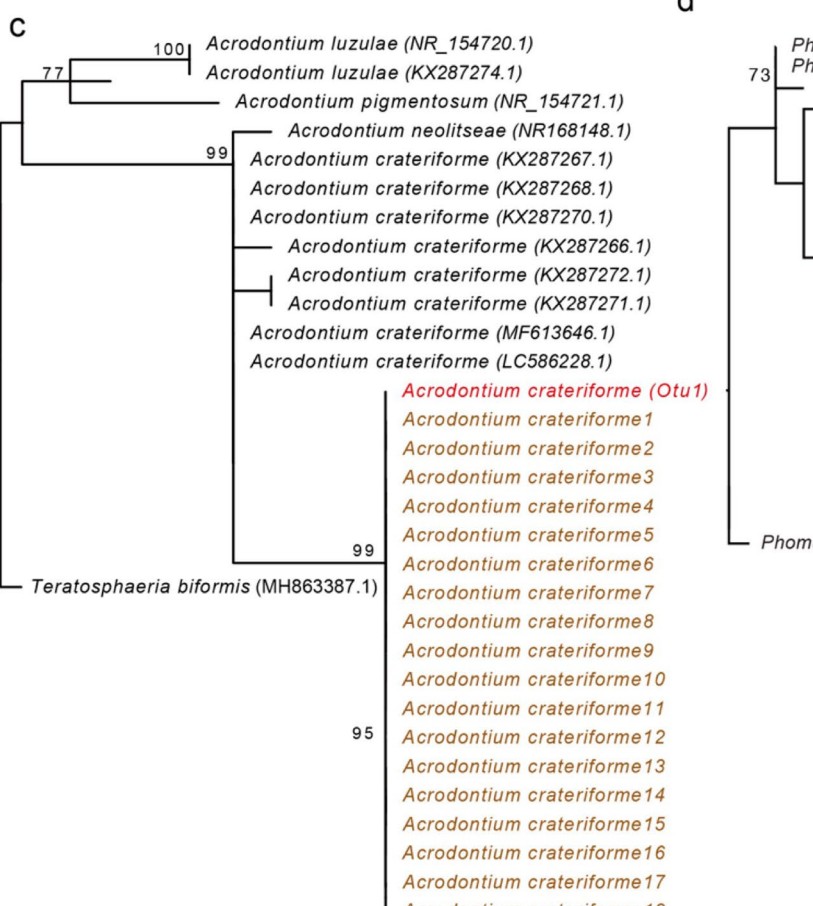

**Extended Data Fig. 2 | ITS phylogeny of *Acrodontium crateriforme* and *Phoma herbarum*.** Photo shows the morphology of **a**. *A. crateriforme* and **b**. *P. herbarum* grown in potato dextrose agar. Red colour represents the consensus sequence of the **c**. *Acrodontium* and **d**. *Phoma* OTU from amplicon data. Brown colour represent sequences of strains that were isolated from *Drosera spatulata* mucilage.

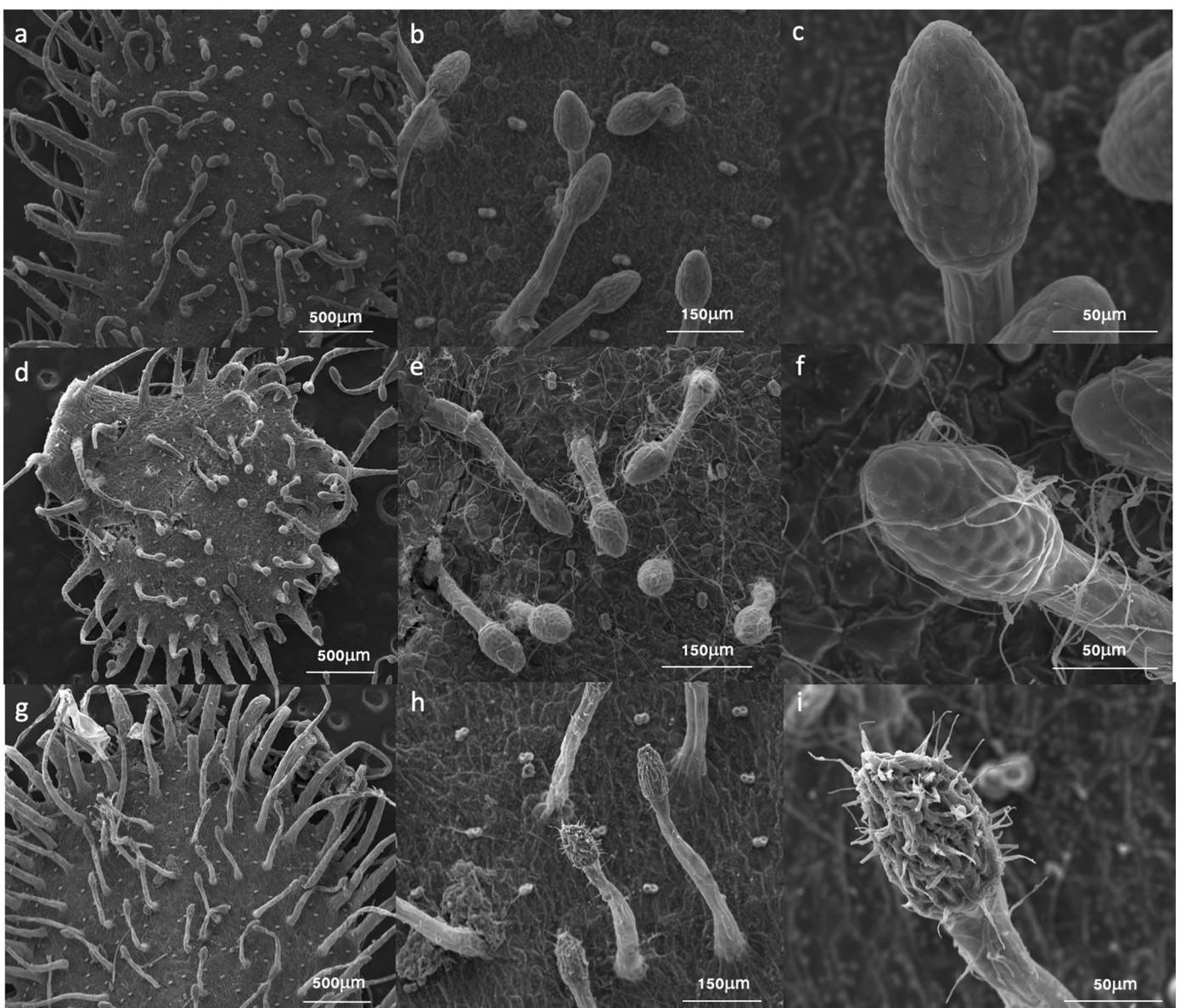

**Extended Data Fig. 3 | Scanning electron microscope (SEM) image of *Drosera spatulata* sundew leaves. a-c**. laboratory material grown under sterile, axenic conditions without fungi or bacteria, **d-f**. laboratory materials inoculated with *A. crateriforme* fungus, and **g-i**. collected from wild. Each condition was repeated once and images were re-taken with similar results.

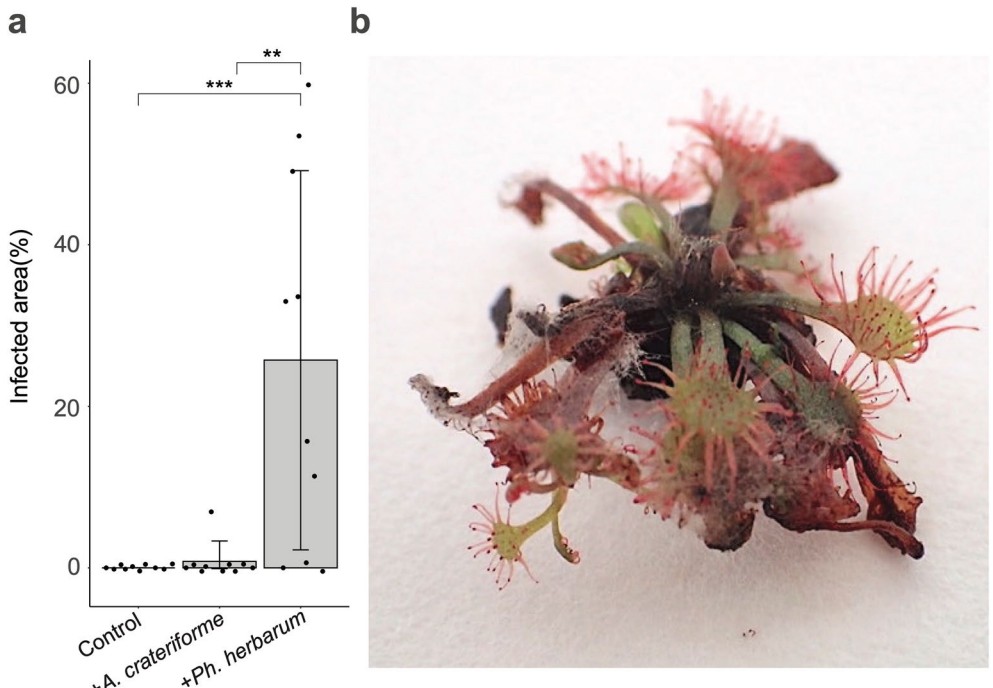

**Extended Data Fig. 4 | Infected areas of *D. spatulata* one month after post-inoculation with *A. crateriforme* or *Ph. herbarum*. a**. Each set consist of wilted area calculated from 10 plants (Wilcoxon rank sum test; two sided, *P < 0.05. **P < 0.01, ***P < 0.001). Data are presented as mean values ± SD. **b**. A photo showing wilt of *D. spatulata* as a result of inoculating *Ph. herbarum*.

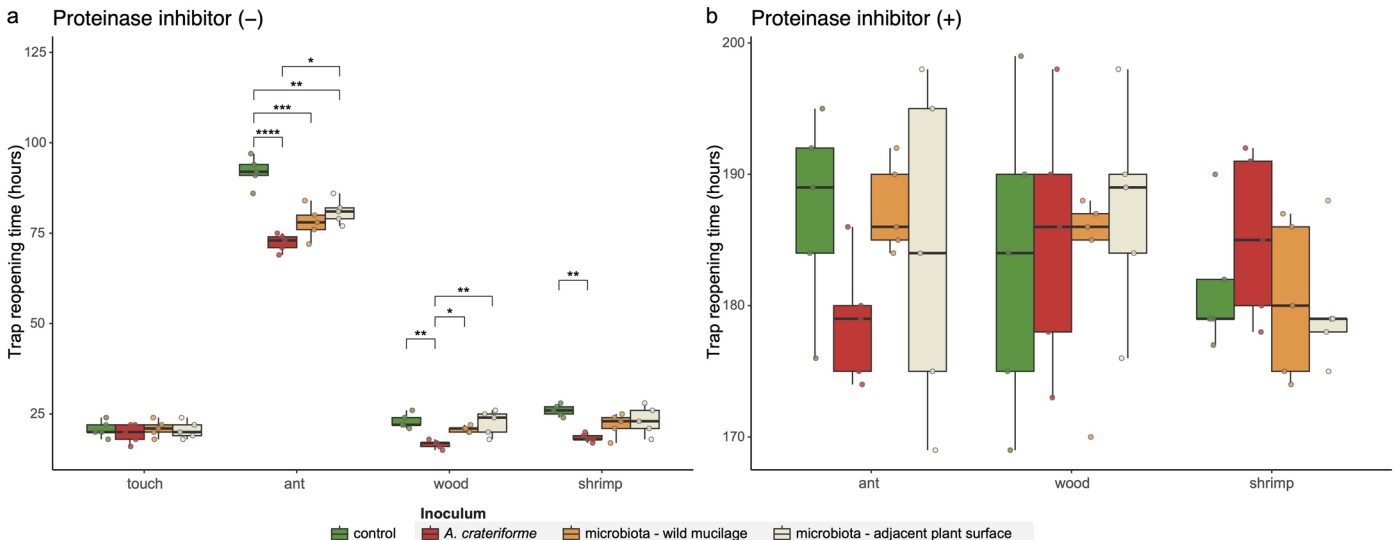

**Extended Data Fig. 5 | Re-opening time of sundew traps grown one month after different inoculum and fed with different substrates/stimulation.** **a**. Treatment without proteinase inhibitor. **b**. Treatment with proteinase inhibitor. Asterisk denote P values from Wilcoxon-rank sum test (two sided, $* P<0.05, ** P<0.01, *** P<0.001$). + and − denote presence and absence of treatment, respectively. The centre line represents the median, with the upper and lower bounds of the box representing the 25th and 75th percentiles, respectively. The whiskers extend to 1.5 × i.q.r.

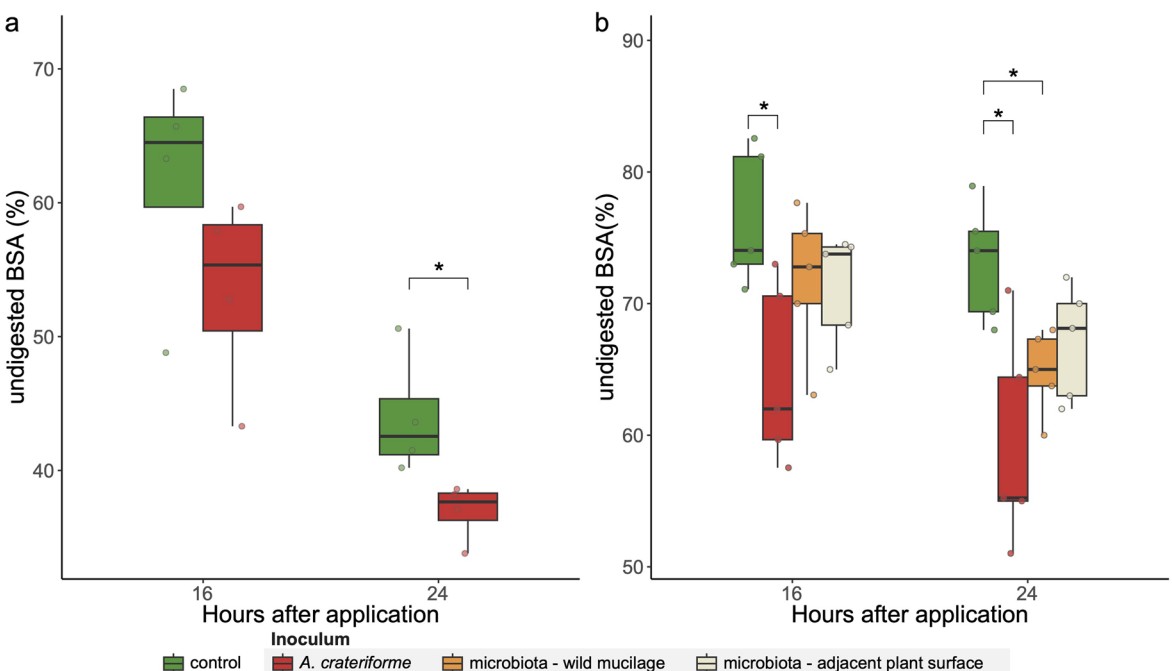

**Extended Data Fig. 6 | Another two batches (a. and b.) of digestion experiment.** Application of biotin-labelled BSA as a protein substrate during 16 and 24 h of sundew digestion using collected mucilage from sundews inoculated with different inoculum. (**a**: n=3, **b**: n=5, biological measurements; Wilcoxon rank sum test with multiple testing; two sided, * adjusted P<0.05.) The centre line represents the median, with the upper and lower bounds of the box representing the 25th and 75th percentiles, respectively. The whiskers extend to 1.5 × i.q.r.

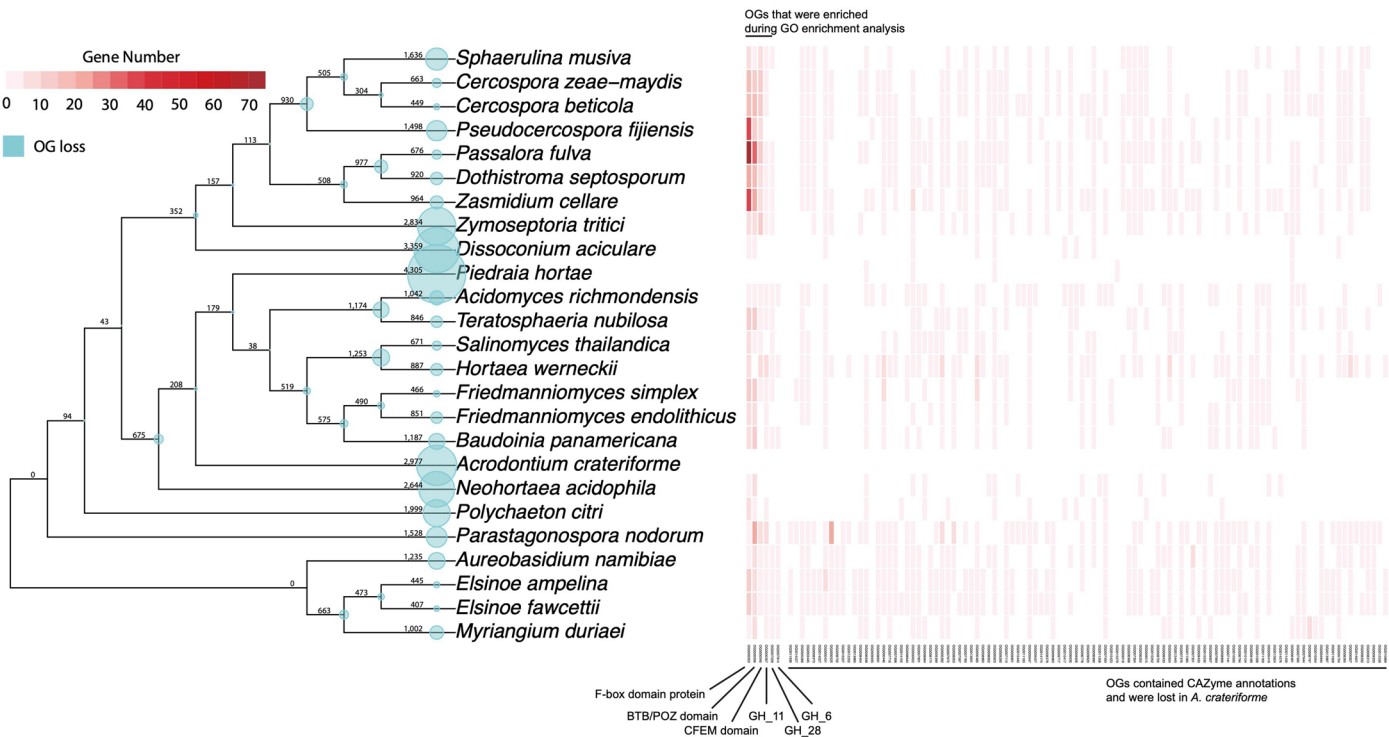

**Extended Data Fig. 7 | *Acrodontium* phylogeny with orthogroup (OG) losses and gene number.** The blue circles on phylogeny described the loss OG number at each node. The heatmaps showed gene numbers of OGs in representative species. *A. crateriforme* was inferred to loss these OGs.

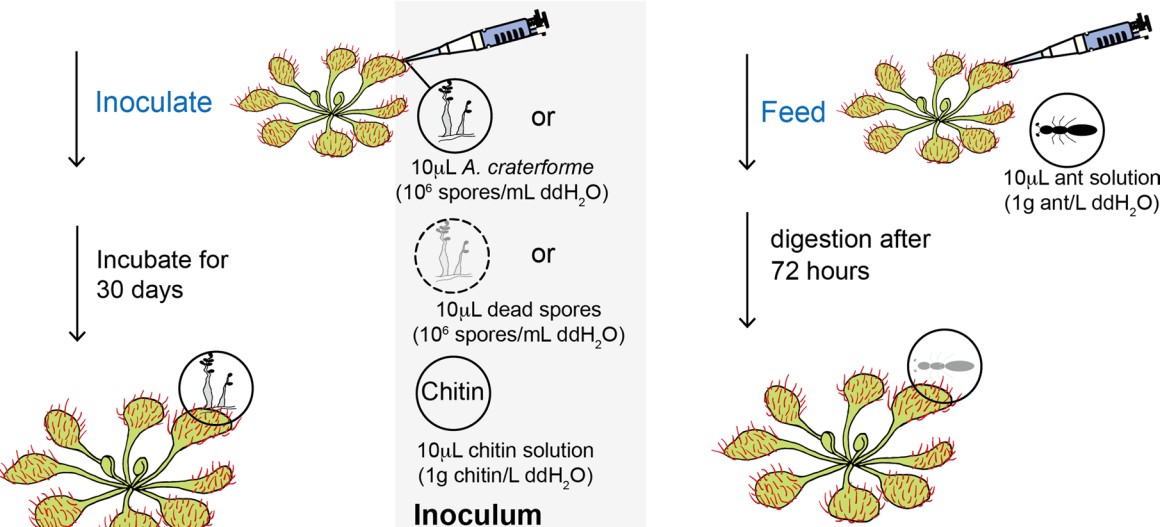

**1. Sterile plants (baseline)**

Inoculate

Incubate for 30 days

10μL *A. craterforme* (10⁶ spores/mL ddH₂O)

or

10μL dead spores (10⁶ spores/mL ddH₂O)

or

Chitin

10μL chitin solution (1g chitin/L ddH₂O)

**Inoculum**

Feed

10μL ant solution (1g ant/L ddH₂O)

digestion after 72 hours

**3. Digestion phase**

**2. Coexistence phase or 4. plants' response to chitin/dead fungus**

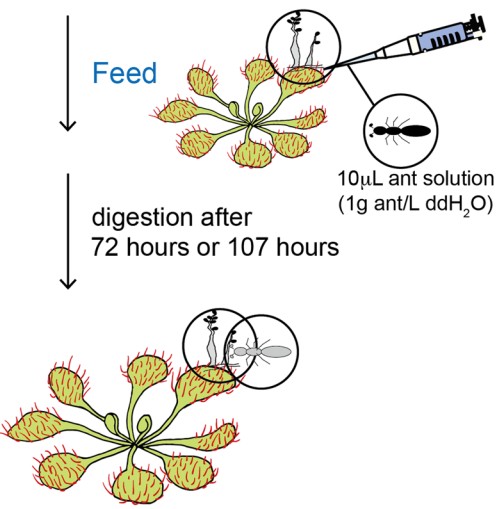

Feed

10μL ant solution (1g ant/L ddH₂O)

digestion after 72 hours or 107 hours

**5. Holobiont digestion**

**Extended Data Fig. 8 | Schematic diagram of different treatments used in the experiment for RNAseq.** Green numbers and letters denote different treatments and transcriptome comparisons. Blue letters denote treatment. Details of preparation of sterile plants and incubation after inoculation are shown in Methods and Supplementary Methods.

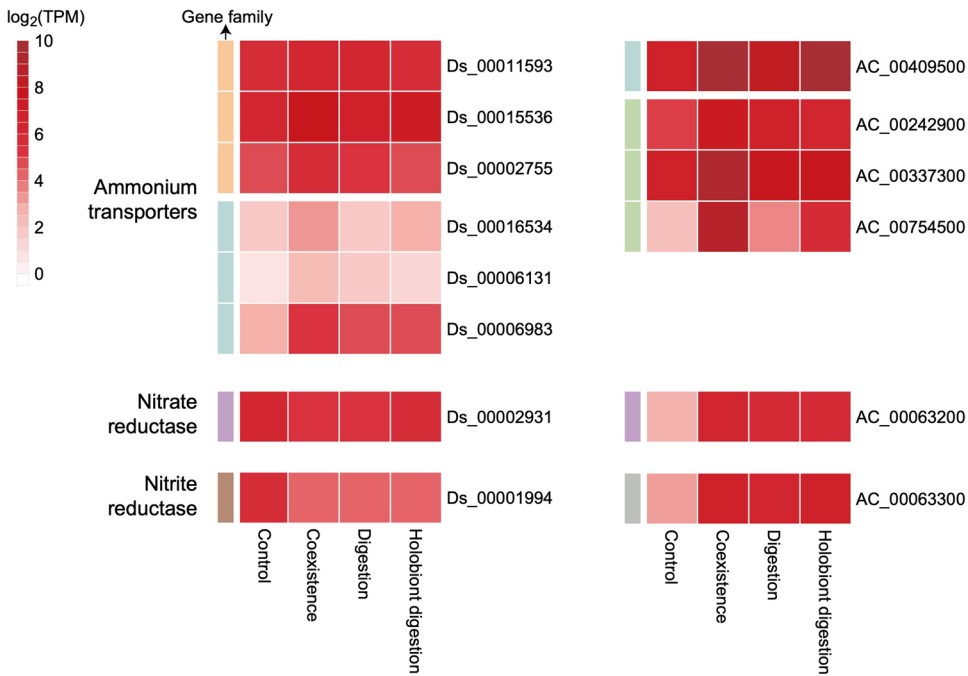

**Extended Data Fig. 9 | Fungal growth on remains of dead arthropod (covered by evenly spread hairs) on wild *Drosera spatulata*.** Fungal hyphae indicated by white arrowheads, two conidiophores of *Acrodontium crateriforme* by red arrows.

**Extended Data Table 1 | References of metabarcoding studies in carnivorous plants**

| Carnivorous plant species | Authors | year | 16S[yes\|no] | 18S or ITS [yes\|no] | dominant bacterial OTUs [yes\|no] | dominant eukatyotic OTUs [yes\|no] | Common bacterial OTUs | Common eukaryotic OTUs | Reanalysis and relative abundance of *Acrodontium* |
|---|---|---|---|---|---|---|---|---|---|
| *Cephalotus follicularis, Nepenthes mirabilis, Nepenthes mindanaoensis* | LS Bittleston et al | 2022 | yes | yes | no | no | no | no | 0 |
| *Darlingtonia californica* | DW Armitage | 2017 | yes | no | yes | NA | *Pseudomonas* sp. 22.77%, *Pedobacter* sp. 10.94% | NA | NA |
| *Dionaea muscipula* | W Sickel et al | 2019 | yes | no | yes | NA | *Mycoplasma* (OTU1, 7.96%), *Acidisoma* (OTU3, 13.4%), *Sphingomonas* (OTU13, 8.6%) | NA | NA |
| *N. albomarginata, N. ampullaria, N. hirsuta, and N. mirabilis* | Y Takeuchi et al | 2016 | yes | no | yes | NA | *N. albomarginata*: *Acinetobacter* 17.3%, *Pseudomonas* 7.7% ; *N. ampullaria*: *Acinetobacter* 14.44%; *N. hirsuta*: *Acinetobacter* 12.66%, *Pseudomonas* 7.8% ; *N. mirabilis*: *Acinetobacter* 10.39%, | NA | NA |
| *N. gracilis, N. rafflesiana, N. ampullaria* | LS Bittleston et al | 2023 | yes | no | yes in *N. gracilis* | NA | *N. gracilis*: *Acidocella* (9.5%), *Acidocella* (5.86%) | NA | NA |
| *N. hemsleyana, N. rafflesiana* | LD Alcaraz et al | 2016 | yes | no | NA | NA | NA | NA | NA |
| *N. rafflesiana, N. hemsleyana* | W Sickel et al | 2016 | yes | no | yes | NA | *N. hemsleyana*: *Klebsiella* (22%), *N. rafflesiana*: *Acidocella* (25%) | NA | NA |
| *Nepenthes stenophylla, Nepenthes veitchii, Nepenthes reinwardtiana, Nepenthes tentaculata, Nepenthes hirsuta, Sarracenia alata, Sarracenia flava, Sarracenia leucophylla, Sarracenia purpurea, Sarracenia rosea* | LS Bittleston et al | 2018 | yes | yes | yes | Y | *S. rosea*: *Aquitalea* (16.73%), *Rhodopseudomonas palustris* (7.1%) ; *S. flava*: *Rhodanobacter* sp. (8.4%) ; *S. leucophylla*: *Serratia* sp. (9.8%) | no | 0 |
| *Sarracenia alata* | MM Koopman and BC Carstens | 2011 | yes | no | yes | NA | Enterobacteriaceae, Comamondaceae, Pseudomonadaceae | NA | NA |
| *Sarracenia minor, Sarracenia flava* | SM Yourstone et al | 2021 | yes | no | NA | NA | no | NA | NA |
| *Sarracenia purpurea* | JJ Grothjan and E Young | 2022 | yes | no | no | NA | no | NA | NA |
| *Sarracenia purpurea* | JJ Grothjan and E Young | 2019 | yes | yes | yes | yes | *Azospirillum* (9.33%), *Pedobacter* (5.85%) | no | 0 |
| *Sarracenia purpurea* | PJ Boynston et al | 2019 | yes | yes | no | yes | no | *Candida pseudoglaebosa* (41%); Present in 9/53 samples | **0.16-1.4% (Present in 4/53 samples)** |
| *U.vulgaris, U.australis* | D Sirova et al | 2018 | yes | yes | no | no | no | NA | 0 |

The dominant/diverse column indicate whether the authors have reported presence of dominant OTUs. Common OTUs were inferred by reanalysing the OTU tables and inferred as having greater than 5% relative abundance. NA denote data not available.

# Reporting Summary

## Statistics

For all statistical analyses, confirm that the following items are present in the figure legend, table legend, main text, or Methods section.

| n/a | Confirmed | |
|---|---|---|
| ☐ | ☒ | The exact sample size (*n*) for each experimental group/condition, given as a discrete number and unit of measurement |
| ☐ | ☒ | A statement on whether measurements were taken from distinct samples or whether the same sample was measured repeatedly |
| ☐ | ☒ | The statistical test(s) used AND whether they are one- or two-sided *Only common tests should be described solely by name; describe more complex techniques in the Methods section.* |
| ☒ | ☐ | A description of all covariates tested |
| ☒ | ☐ | A description of any assumptions or corrections, such as tests of normality and adjustment for multiple comparisons |
| ☐ | ☒ | A full description of the statistical parameters including central tendency (e.g. means) or other basic estimates (e.g. regression coefficient) AND variation (e.g. standard deviation) or associated estimates of uncertainty (e.g. confidence intervals) |
| ☐ | ☒ | For null hypothesis testing, the test statistic (e.g. *F*, *t*, *r*) with confidence intervals, effect sizes, degrees of freedom and *P* value noted *Give P values as exact values whenever suitable.* |
| ☒ | ☐ | For Bayesian analysis, information on the choice of priors and Markov chain Monte Carlo settings |
| ☒ | ☐ | For hierarchical and complex designs, identification of the appropriate level for tests and full reporting of outcomes |
| ☒ | ☐ | Estimates of effect sizes (e.g. Cohen's *d*, Pearson's *r*), indicating how they were calculated |

*Our web collection on statistics for biologists contains articles on many of the points above.*

## Software and code

Policy information about availability of computer code

Data collection
32 genomes from representative fungi and plants were downloaded from JGI and NCBI databases
https://mycocosm.jgi.doe.gov/Sepmu1/Sepmu1.home.html
https://mycocosm.jgi.doe.gov/Cerzm1/Cerzm1.home.html
https://www.ncbi.nlm.nih.gov/datasets/genome/GCF_002742065.1/
https://mycocosm.jgi.doe.gov/Mycfi2/Mycfi2.home.html
https://mycocosm.jgi.doe.gov/Clafu1/Clafu1.home.html
https://mycocosm.jgi.doe.gov/Dotse1/Dotse1.home.html
https://mycocosm.jgi.doe.gov/Zasce1/Zasce1.home.html
https://mycocosm.jgi.doe.gov/Mycgr3/Mycgr3.home.html
https://mycocosm.jgi.doe.gov/Disac1/Disac1.home.html
https://mycocosm.jgi.doe.gov/Pieho1_1/Pieho1_1.home.html
https://mycocosm.jgi.doe.gov/Aciri1_iso/Aciri1_iso.home.html
https://mycocosm.jgi.doe.gov/Ternu1/Ternu1.home.html
https://mycocosm.jgi.doe.gov/Horth1/Horth1.home.html
https://mycocosm.jgi.doe.gov/Horwer1/Horwer1.home.html
https://www.ncbi.nlm.nih.gov/datasets/genome/GCA_005059865.1/
https://www.ncbi.nlm.nih.gov/datasets/genome/GCA_030411845.1/
https://mycocosm.jgi.doe.gov/Bauco1/Bauco1.home.html
https://mycocosm.jgi.doe.gov/Horac1/Horac1.home.html
https://mycocosm.jgi.doe.gov/Polci1/Polci1.home.html
https://mycocosm.jgi.doe.gov/Aurpu_var_nam1/Aurpu_var_nam1.home.html

https://mycocosm.jgi.doe.gov/Elsamp1/Elsamp1.home.html
https://www.ncbi.nlm.nih.gov/datasets/genome/GCA_012977835.1/
https://mycocosm.jgi.doe.gov/Myrdu1/Myrdu1.home.html
https://www.ncbi.nlm.nih.gov/datasets/genome/GCF_000146915.1/
SILVA database (v138.1) and the UNITE90 database (v9)
Uniprot Fungi (version October 2019)

Data analysis

Comparative genomics and phylogenomics were using: Orthofinder (ver. 2.5.5), mafft (version 7.741), IQtree (version 2.2.2.6), ASTRAL-III (ver. 5.7.1) and DOLLOP (ver. 3.69.650). Transcriptome analysis were performed using fastp (v0.23.2), STAR (v 2.7.10b), featureCounts (v 2.0.3), DESeq2 (v1.38.3), and topGO (v2.50.0). Genome sequencing, assembly and annotation of A. crateriforme were performed using the canu assembler (ver. 1.9), the flye assembler (ver. 2.5), Racon (four iterations; ver. 1.4.11), Medaka (ver. 0.11.0), Pilon(ver. 1.24) and NOVOPlasty (ver. NOVOPlasty2.7.O.pl). Gene prediction and annotation were performed using open-source tools: STAR (ver. 2.7.7a), Trinity (ver 2.13.2), Trinity (ver 2.13.2; guided approach), Stringtie (ver 2.1.7), Cufflinks (ver2.2.1), Minimap2 (ver 2.1), Portcullis (ver 1.2.3), Augustus (ver 3.4.0), BRAKER2 (ver. 2.1.6), SNAP (ver 2006-07-28), MIKADO (ver 2.3.3), Uniprot Fungi (version October 2019), MAKER2 (ver 3.01.03), Repeatmasker (ver 4.1.2). pfam_scan (ver. 1.6), Pfam database (ver. 36), Diamond (ver. 2.1.6), TransportDB (ver. 2.0), dbCAN (ver. 2.0.11), MEROPS (ver. 12.4), antiSMASH (fungi ver. 7.0.1) and eggNOG (ver. 2.1.12). Amplicon samples were demultiplexed using sabre (v1.0), USEARCH (v11.0.667), phyloseq package (v1.46.0), SILVA database (v138.1) and UNITE database (v.9), rgbif package (v3.7.7).

Analysis and visualisations were conducted under R environment (ver. 4.3.1). Various packages were utilised: topGO79 (ver. 2.52.0), pheatmap (ver. 1.0.12) and ggplot2122 (ver. 3.4.4). The amount of protein remaining and the growth area of A. crateriforme were measured using ImageJ (v 1.53).

For manuscripts utilizing custom algorithms or software that are central to the research but not yet described in published literature, software must be made available to editors and reviewers. We strongly encourage code deposition in a community repository (e.g. GitHub). See the Nature Portfolio guidelines for submitting code & software for further information.

# Data

Policy information about availability of data

All manuscripts must include a data availability statement. This statement should provide the following information, where applicable:

- Accession codes, unique identifiers, or web links for publicly available datasets
- A description of any restrictions on data availability
- For clinical datasets or third party data, please ensure that the statement adheres to our policy

All sequences generated from this study were deposited on NCBI under BioProject PRJNA1034788 and PRJNA1095839. The genome and annotation of A. crateriforme is available in NBCI under accession GCA_033807595.1
Accession numbers of individual samples can be found in Supplementary Table 1 and 11.

# Research involving human participants, their data, or biological material

Policy information about studies with human participants or human data. See also policy information about sex, gender (identity/presentation), and sexual orientation and race, ethnicity and racism.

| | |
|---|---|
| Reporting on sex and gender | N/A |
| Reporting on race, ethnicity, or other socially relevant groupings | N/A |
| Population characteristics | N/A |
| Recruitment | N/A |
| Ethics oversight | N/A |

Note that full information on the approval of the study protocol must also be provided in the manuscript.

# Field-specific reporting

Please select the one below that is the best fit for your research. If you are not sure, read the appropriate sections before making your selection.

☐ Life sciences    ☐ Behavioural & social sciences    ☒ Ecological, evolutionary & environmental sciences

For a reference copy of the document with all sections, see nature.com/documents/nr-reporting-summary-flat.pdf

# Ecological, evolutionary & environmental sciences study design

All studies must disclose on these points even when the disclosure is negative.

| | |
|---|---|
| Study description | We collected D. spatulata mucilage samples from five collection sites located in Northern Taiwan. We also collected multiple Drosera species in the UK and USA |
| Research sample | Drosera spatulata mucilage and co-occurring plants; multiple Drosera tissues |
| Sampling strategy | *Note the sampling procedure. Describe the statistical methods that were used to predetermine sample size OR if no sample-size calculation was performed, describe how sample sizes were chosen and provide a rationale for why these sample sizes are sufficient.* |
| Data collection | The fresh leaves of plants and mosses surrounding D. spatulata were also wiped and considered as environmental samples. Each sample is a pooled mucilage from 30 D. spatulata leaves of the same site by using filter paper of size 1 cm x 1.5 cm. |
| Timing and spatial scale | Sample collecting date is started from June 7th, 2018 to July 28th, 2021. In temporal experiment, we collected sample from June 7th, 2018 to April 23th. we collect samples once a month in Sumei (25.1014697568718N, 121.860991424351E) and Shuangxi (25.066234N, 121.843341E). In spatial experiment, we collect 18 different location (Buyanting: 25.090083N,121.847242E; Houtong: 25.086944N, 121.827778E; Keelung1: 25.159125N, 121.704949E; Keelung2: 25.155061N, 121.710643E, Keelung3: 25.158397N, 121.709447E; Nangang: 25.047277N, 121.639736E;  Pingxi1: 25.06611N, 121.71027E; Pingxi2: 25.061111N, 121.78554E; Pingxi3: 25.07416N, 121.79306E; Pingxi4: 24.992262N, 121.663487E; Shiding1: 24.981645N, 121.614058E; Shiding2: 24.97312N, 121.615112E; Shiding3: 24.954655N, 121.624628E; Shuangxi2: 24.970089N, 121.825341E; Shuangxi3: 24.937508N ,121.837337E; Yilan1: 24.85931N, 121.77751E; Yilan2: 24.84593N, 121.77647E) from July 6th, 2021 to July 28th 2021. To compare D. spatulata mucilage and the adjacent environment, we collect samples in Shuangxi(25.066234N, 121.843341E), Shumei(25.1014697568718N, 121.860991424351E) and Yangmingshan(25.181007N, 121.555332E; 25.1769170520255N, 121.558009143632E; 25.1800240781841N, 121.549426075361E) during July, 2019. |
| Data exclusions | Two ITS rRNA gene amplicon samples were excluded from our analyses due to low read numbers. |
| Reproducibility | N/A for community experiment |
| Randomization | D. spatulata mucilage samples and the adjacent environmental samples were randomly collected into the experimental groups |
| Blinding | Blind testing was not directly relevant to the experiment. |

Did the study involve field work?   ☒ Yes   ☐ No

## Field work, collection and transport

| | |
|---|---|
| Field conditions | The research was studied between June, 2018 to April ,2019. |
| Location | Drosera spatulata were located in Sumei (25.1014697568718N, 121.860991424351E; 24.970089N, 121.825341E; 24.937508N ,121.837337E), Shuangxi (25.066234N, 121.843341E), Buyanting (25.090083N,121.847242E), Houtong (25.086944N, 121.827778E), Keelung (25.159125N, 121.704949E;  25.155061N, 121.710643E; 25.158397N, 121.709447E), Nangang (25.047277N, 121.639736E), Pingxi (25.06611N, 121.71027E;  25.061111N, 121.78554E; 25.07416N, 121.79306E; 24.992262N, 121.663487E), Shiding (24.981645N, 121.614058E; 24.97312N, 121.615112E; 24.954655N, 121.624628E) Yilan (24.85931N, 121.77751E; 24.84593N, 121.77647E)  and Yangmingshan(25.181007N, 121.555332E; 25.1769170520255N, 121.558009143632E; 25.1800240781841N, 121.549426075361E). |
| Access & import/export | The fresh leaves of plants and mosses surrounding D. spatulata were also wiped and considered as environmental samples. Each sample is a pooled mucilage from 30 D. spatulata leaves of the same site by using filter paper of size 1 cm x 1.5 cm. |
| Disturbance | The disturbance of the environment by sampling was minimal. |

# Reporting for specific materials, systems and methods

We require information from authors about some types of materials, experimental systems and methods used in many studies. Here, indicate whether each material, system or method listed is relevant to your study. If you are not sure if a list item applies to your research, read the appropriate section before selecting a response.

## Materials & experimental systems

| n/a | Involved in the study |
|---|---|
| ☐ | ☒ Antibodies |
| ☒ | ☐ Eukaryotic cell lines |
| ☒ | ☐ Palaeontology and archaeology |
| ☒ | ☐ Animals and other organisms |
| ☒ | ☐ Clinical data |
| ☒ | ☐ Dual use research of concern |
| ☐ | ☒ Plants |

## Methods

| n/a | Involved in the study |
|---|---|
| ☒ | ☐ ChIP-seq |
| ☒ | ☐ Flow cytometry |
| ☒ | ☐ MRI-based neuroimaging |

# Antibodies

| | |
|---|---|
| Antibodies used | Horseradish peroxidase (HRP)-conjugated streptavidin (Cat No. SA00001-0) Dilution rate: (1:5000). |
| Validation | Streptavidin for biotin. Applications for ELISA, WB, Dot blot. |

# Dual use research of concern

Policy information about dual use research of concern

## Hazards

Could the accidental, deliberate or reckless misuse of agents or technologies generated in the work, or the application of information presented in the manuscript, pose a threat to:

| No | Yes | |
|---|---|---|
| ☒ | ☐ | Public health |
| ☒ | ☐ | National security |
| ☒ | ☐ | Crops and/or livestock |
| ☒ | ☐ | Ecosystems |
| ☒ | ☐ | Any other significant area |

## Experiments of concern

Does the work involve any of these experiments of concern:

| No | Yes | |
|---|---|---|
| ☒ | ☐ | Demonstrate how to render a vaccine ineffective |
| ☒ | ☐ | Confer resistance to therapeutically useful antibiotics or antiviral agents |
| ☒ | ☐ | Enhance the virulence of a pathogen or render a nonpathogen virulent |
| ☒ | ☐ | Increase transmissibility of a pathogen |
| ☒ | ☐ | Alter the host range of a pathogen |
| ☒ | ☐ | Enable evasion of diagnostic/detection modalities |
| ☒ | ☐ | Enable the weaponization of a biological agent or toxin |
| ☒ | ☐ | Any other potentially harmful combination of experiments and agents |

## Plants

| | |
|---|---|
| Seed stocks | drosera spatulata was collected in Yangmingshan(25.1769170520255N, 121.558009143632E) in July, 2018. seeds were sterilised by 70% ethanol and 3% (w/v) calcium hypochlorite (CaCl2O2). Surface sterilised seeds were pregerminated on 0.5% water agar at 20°C for 6-8 weeks. Shoots were then transferred to 1/2 MS agar with pH adjusted to 5.7. |
| Novel plant genotypes | No novel plant genotypes in this manuscript |
| Authentication | No authentication in this manuscript |

