## [Peer Review File · Nature Microbiology]

Peer Review Information

Journal: Nature Microbiology

Manuscript Title: An acidophilic fungus promotes prey digestion in a carnivorous plant

Corresponding author name(s): Dr Isheng Tsai

Reviewer Comments & Decisions:

Decision Letter, initial version:

Message: 17th January 2024

Dear Jason,

Thank you for your patience while your manuscript "An acidophilic fungus is integral to prey digestion in a carnivorous plant" was under peer-review at Nature Microbiology. It has now been seen by 4 referees, whose expertise and comments you will find at the end of this email. Although they find your work of some potential interest, they have raised a number of concerns that will need to be addressed before we can consider publication of the work in Nature Microbiology.

In particular, referee #1 is concerned that the findings could also be due to Drosera traps and fungus competing for food and the traps opening earlier. The referee suggests to perform experiments that compares responses to intact fungus, dead fungus or just plain chitin samples. The referee also says that the transcriptome sequencing results should be compared to those from a previous study. Furthermore, referee #1 asks to describe in more detail the experimental procedures, including the application of ant powder onto single leaflets. The referee also feels that the RNASeq analyses need to be better described, and the referee has some technical concerns on how the transcriptomic dataset was analysed. Referee #1 also suggests to add an *A. crateriforme*-inoculated sample treated with prey to test for an additional boost of JA induction in the presence of the assumed fungal interaction partner. Editorially, we will require you to address these concerns in full. Referee #4 is concerned that the samples were collected from a geographically-narrow area and says it remains unclear whether this association remains stable across larger geographical scales. We encourage you to address this point if possible, but won't make it a requirement for publication. Referee #4 says that the experimental evidence demonstrating that the fungus is needed for efficient prey digestion is not very strong and suggests using a negative control such as inert powder or powder from another (less digestible, non insect) organism that is not a natural sundew's prey. This will need to be addressed. Further, the referee suggests to consider microbiota transplantation from natural plants to sterile plants and test whether prey digestion with *D. spatulata* alone recapitulates prey digestion of plants recolonized with more complex mucilage microbiomes. Referee #4 also states that the study does not provide genetic evidence that fungal genes encoding aspartic proteases are actually contributing to prey digestion. Editorially, we encourage you to test this if possible, but this won't be a requirement for publication. The remaining concerns however should be addressed in full.

2Should further experimental data allow you to address these criticisms, we would be happy to look at a revised manuscript.

Please include a data availability statement as a separate section after Methods but before references, under the heading "Data Availability". This section should inform readers about the availability of the data used to support the conclusions of your study. This information includes accession codes to public repositories (data banks for protein, DNA or RNA sequences, microarray, proteomics data etc...), references to source data published alongside the paper, unique identifiers such as URLs to data repository entries, or data set DOIs, and any other statement about data availability. At a minimum, you should include the following statement: "The data that support the findings of this study are available from the corresponding author upon request", mentioning any restrictions on availability. If DOIs are provided, we also strongly encourage including these in the Reference list (authors, title, publisher (repository name), identifier, year). For more guidance on how to write this section please see: <http://www.nature.com/authors/policies/data/data-availability-statements-data-citations.pdf>

- * If you have not done so already we suggest that you begin to revise your manuscript so that it conforms to our Article format instructions at <http://www.nature.com/nmicrobiol/info/final-submission>. Refer also to any guidelines provided in this letter.

2When submitting the revised version of your manuscript, please pay close attention to our [href="https://www.nature.com/nature-portfolio/editorial-policies/image-integrity">Digital Image Integrity Guidelines](https://www.nature.com/nature-portfolio/editorial-policies/image-integrity). and to the following points below:

Note: This url links to your confidential homepage and associated information about manuscripts you may have submitted or be reviewing for us. If you wish to forward this e-mail to co-authors, please delete this link to your homepage first.

Nature Microbiology is committed to improving transparency in authorship. As part of our efforts in this direction, we are now requesting that all authors identified as 'corresponding author' on published papers create and link their Open Researcher and Contributor Identifier (ORCID) with their account on the Manuscript Tracking System (MTS), prior to acceptance. This applies to primary research papers only. ORCID helps the scientific community achieve unambiguous attribution of all scholarly contributions. You can create and link your ORCID from the home page of the MTS by clicking on 'Modify my Springer Nature account'. For more information please visit please visit www.springernature.com/orcid.

If you wish to submit a suitably revised manuscript we would hope to receive it within 6 months. If you cannot send it within this time, please let us know. We will be happy to consider your revision, even if a similar study has been accepted for publication at Nature Microbiology or published elsewhere (up to a maximum of 6 months).

Reviewer Expertise:

Referee #1: Carnivorous plants

3Referee #2: Plant mucilage microbiome

Referee #3: Evolution of host-fungal symbioses

Referee #4: Plant-microbe interactions, genomics, transcriptomics, 16S

Reviewer Comments:

Reviewer #1 (Remarks to the Author):

In their study by Sun et al aimed to prove the assumption that symbiotic interaction between the sundew *Drosera spatulata* and the acidophilic fungus *Acrodonium crateriforme* is crucial for prey digestion and thus *Drosera*'s carnivorous lifestyle. To this end, the authors have merged data derived from a set of individual experiments to support above mentioned symbiotic interaction hypothesis. While the bacterial community found in *Drosera*'s mucilage resembled that of the surrounding plants, the fungal community was significantly different, with *Acrodonium crateriforme* representing the dominant fungal species regardless of sampling site or season. Based on the observation that duration of the digestive process is shortened in traps inoculated with *A. crateriforme*, the authors suggest that the presence of this acidophilic fungus enhanced digestion, thus allowing the plant to effectively consume more prey.

General comment

The reader may want to learn about the microbial situation in *Dionaea*. Thus, the authors may want to cite and discuss:

Venus flytrap carnivorous lifestyle builds on herbivore defense strategies.

Bemm et al. *Genome Res.* 2016 Jun;26(6):812-25.

Venus flytrap microbiotas withstand harsh conditions during prey digestion. Sickel et al. 2019 *FEMS Microbiol Ecol.* Mar 1;95(3). doi: 10.1093/femsec/fiz010. PMID: 30649283.

Questions:

Q1: From their results the authors conclude that *A. crateriforme* enhanced the digestion process of the trap. This could also be interpreted in terms of trap reopening under the control of the that is actually size of food stock present. In other words, when *Drosera* traps and fungus compete for food, the food stock is depleted earlier and thus the trap opens early. A notion that seems supported by the authors own finding 'The *A. crateriforme* ammonium transporters as well as nitrate reductase also exhibited the same expression trends (Supplementary Fig. 22), suggesting active ammonium exchange and utilisation already taking place within the plant-fungus holobiont.'

Q2: The authors mention 'The *D. spatulata* DEGs had a higher proportion of multi-function and additive DEGs than the fungus, consistent with the optimisation of the genes repurposed to involve digestion while the majority still retained the ancestral function of species interaction.'

The reader does not quite understand this issue and suggests two perform experiments that compares responses to intact fungus, dead fungus or just plain chitin samples. From

4this experiment one will learn, whether or not the presence of the elicitors such as chitin are essential and sufficient to trigger *D. spatulata* DEGs, as it is the situation in *Dionaea* (Bemm et al. 2016).

Q3: The authors mention that additional transcriptome sequencing was performed in *D. spatulata* towards the end of digestion process. Concerning this issue the authors may wish to relate their findings to those of a related study on *Dionaea* (Rentsch et al, 2023. Comparative transcriptomics of Venus flytrap (*Dionaea muscipula*) across stages of prey capture and digestion.)

Q4: The reader is wondering how one can apply "10-5g of sterilized ant powder ...on a single leaf..." (line 686) for the feeding experiments – and appreciates a more detailed description of how to apply such a tiny amount of prey powder in a controlled fashion. The method part should contain information on how the authors assure that the weighed protein amount reaches the proper trap regions based on proper statistics? In addition to adding minute amounts of prey powder to *Drosera* traps, feeding experiments must be performed with intact living or dead prey with the responses compared.

Q5: In lines 271-274, the authors state that "A. crateriforme growth is positively correlated with the amount of ant powder to the medium, suggesting that the fungus can utilize insects as growth supplement.", relating to supplementary Fig. 10. The data shown there, however, is not *A. crateriforme* growth after addition of various amounts of ant powder. Instead, 1g of ant powder is added to three different media. As far as I can see, growth area is smallest (!) in each of the three media containing 1g of ant powder compared to control or dextrose-supplemented medium. So, there's no positive correlation of fungal growth and ant powder supplementation. As already asked above, the authors might therefore want to reconsider the conclusion that the fungus can benefit from insect digestion for its own growth.

Q6: Genome sequencing detected genomic rearrangements in *A. crateriforme* the authors discuss as adaptation to its acidic environment and symbiotic interaction with plants. RNASeq analyses in untreated controls of both *A. crateriforme* and *D. spatulata* and inoculated and/or fed holobiont samples were performed. This complex experiment is difficult for the reader to understand, partly because of poor description in the supplement (see comment below).

Q7: As far as the reader understands, data derived from inoculated & fed samples (coexistence & digestion samples as well as holobiont) were mapped to both plant and fungus genomes (lines 718 & 719). Then, the authors state that more than 60% of DEGs "identified in the digestion phase for each species were co-expressed in the coexistence phase" (lines 404-406).

Mapping the reads to both genomes, how can one reliably assign a read to the right species? Obviously, reads mapping to both species' genomes were removed (lines 720 & 721) - Moreover, the length of the grey "overlap"-bar in Figure 5a doesn't represent more than 60% of all DEGs, especially for the DEGs downregulated in the fungus. Since this transcriptomic dataset is central for the author's hypothesis, the reader would like to find

5

the experiment(s) and data analysis properly described.

Q8: To the reader it is not a real surprise that DEGs between digesting and inoculated (coexistence) samples overlap and make the authors speculate about co-option of defense-involved genes for prey digestion. Previous related studies on *Drosera* & other carnivorous plants before, came to the same conclusion. The author may want to read cite these papers relevant to the topic discussed here.

Q9: The authors focus on genes that were only DEGs in the presence of both fungus and prey (holobiont) and categorize these based on assignment to the individual experiments described before. They suggest that "more than half of upregulated DEGs in both species were designated having multiple or gene-opted", thus pointing to co-evolution of plant & fungus and optimization of the holobiont transcriptome including synergistic expression of digestive enzymes and nutrient transporters.

Finally, the authors quantify JA levels in differently treated traps, pointing to an increase in JA-Ile after "application of both ant powder and microbes" (line 534 & 535). Unfortunately, there's no *A. crateriforme*-inoculated sample treated with prey to test for an additional boost of JA induction in the presence of the assumed fungal interaction partner.

Q10: Coming back to the transcriptomic data, JA-signalling genes are analyzed under various conditions, leading the authors to conclude that JA signalling genes are coexpressed in both inoculated and ant-fed plants (coexistence & digestion). The heatmap (Fig. 6b) shows log₂(TPM), which makes it hard to see the differences. Many of the genes depicted seem to be comparably expressed under all conditions, even in the control. Maybe giving (log₂)FoldChange would bring some more clarity. It is yet unclear, why the authors choose these data for their final figure, since they do not extensively discuss these results. Thus, one would suggest to keep only Figure 6c and move the JA data to the supplement.

Minor comments:

- Supplementary Methods & Supplementary figure legends contain numerous typos or are difficult to understand. Example: Supplemental Figure 19: "Coexistence samples mean *Acrodontium crateriforme* was inoculated on *Drosera spatulata* for one month. Digestion samples show ant powder add on *Drosera spatulata* for 3 days. Coexistence digestion samples show ant powder add on inoculated *Drosera spatulata* for 3 days. To make sure the impact of RNA expression in late stage, we add ant powder add on *Drosera spatulata* for 107 hours." The last sentence doesn't make much sense. The authors may want to carefully revise the supplementary section.

- Lines 729-731: The authors write "Due to the present (presence??) of peptidase without any expressions across conditions in *D. spatulata*, we removed the 30% lowest-expressed genes in each transcriptome using the sum of samples." What does that mean? How can you justify omission of data based on intricacy to detect certain transcripts?

Reviewer #2 (Remarks to the Author):

This paper identifies *Acrodontium crateriforme* as the dominant fungal constituent of the

6Drosera mucilage engaged in insect capture and digestion. The data nicely demonstrate the role of the fungus in insect and protein digestion and implicate it as the major partner allowing the evolution of insectivory.

Reviewer #3 (Remarks to the Author):

This manuscript describes a very interesting association between a fungus and the carnivorous sundew plant. This work is an excellent investigation and presents a series of future questions to base on using these genomic and analytic analyses.

Overall I find this a well written and analyzed project. The amplicon, culturing, metabolism and comparative analysis provide a clear perspective on this fungal-plant interaction. I find the work fascinating and very interesting exploration of plant-fungal interactions in a really quite different dimension. I appreciate the efforts as well to connect the mycobiome profile to those of pitcher plant work (*Nepenthes* sp) as I would have been curious or expected there to be some comparative analyses done - this is a good context to have in this manuscript.

While not critical -- I do not think the authors have space to add more analyses - but I wonder if the authors had looked at the MATing type locus for its presence in the assembled genome and if the architecture give you any ideas on mating capabilities?

Around line 370 there is discussion of chromosomal synteny but some local synteny which was coined as 'mesosynteny' in Hane et al 2011 <https://link.springer.com/article/10.1186/gb-2011-12-5-r45> -- so not sure if that more general concept is what applies here or if the Capnodiales pattern is distinct as cited for Ohm et al 2012.

Note that *Hortaea acidophila* is now named *Neohortaea acidophila* <https://ncbi.nlm.nih.gov/Taxonomy/Browser/wwwtax.cgi?id=245834> <https://doi.org/10.3767/003158514X681981> so this should be updated in the trees.

Reviewer #4 (Remarks to the Author):

The manuscript by Sun and colleagues is describing a potentially interesting finding, namely that prey digestion in a carnivorous plant is promoted by the presence of an acidophilic fungus. The manuscript has potential but there are numerous important issues that hinder the potential impact of the work.

1) Microbiome profiling

Authors showed interesting microbiome profiling data and convincingly showed that the mucilage microbiome has a distinct fungal microbiome. However, they used the very

7outdated OTU-based clustering method. Nothing against this but authors might have underestimated the true diversity of bacteria and fungi in their samples, including in mucilage.

My major concern is more that the samples were collected from a geographically-narrow area and it remains unclear whether this association remains stable across larger geographical scales. It would be important to test whether *D. spatulata* host similar or dissimilar dominant fungal taxa in at least one completely independent plant population outside Taiwan. Authors actually showed that *A. crateriforme* might not be a dominant microbiota member in other carnivorous plants. In other words, is this potentially beneficial association retained, irrespective of the environment? or can different fungi functionally complement *A. crateriforme* in other environments to promote prey digestion? This information would elevate the potential impact of the work.

2) Experimental evidence

The experimental evidence demonstrating that the fungus is needed for efficient prey digestion is rather weak (Figure 3e,f) and I personally think that these experiments are insufficient to conclude that the fungus is integral to prey digestion. In these experiments, authors prepared sterile carnivorous plants, re-colonized these plants with the fungus and test whether fungal inoculation leads to change in tentacle behavior or promote biotin-labelled BSA breakdown.

The proxy used to determine prey digestion is based on opening and closing time of tentacles. Why using ant powder and why looking at such indirect proxy? Authors might consider a negative control such as inert powder or powder from another (less digestible, non insect) organism that is not a natural sundew's prey. Why not setting up experiments with dead ants and monitor ant biomass degradation based on quantitative measurements? A potential issue is that the presence of the fungus might also modulate tentacle behavior independently from prey digestion.

The experiment with BSA is important because it truly reflects protein degradation capability rather than tentacle behavior. However, one need to realize that the fungal effect appears rather limited in this case. Can the same weak effect be observed with other non-pathogenic fungi? Does this effect depend on the concentration of the fungal inoculum?

The microscopic pictures showed very different colonization patterns between *A. crateriforme* recolonized plants and natural plants, raising the possibility that in nature, *A. crateriforme* is only one, out of many others that potentially contribute to prey digestion in mucilage. Authors could also consider microbiota transplantation from natural plants to sterile plants and test whether prey digestion with *D. spatulata* alone recapitulates prey digestion of plants recolonized with more complex mucilage microbiomes. Is *D. spatulata* really the major player in this process?

The number of replicates (n=5 technical replicates) for these experiments is also questionable. Based on what I can read, these experiments were not even performed three times independently but derived from one experiment with 5 technical replicated. For publication in Nature Microbiology, fully independent replicates should be presented.

3) RNA-Seq

Authors nicely showed that there is a unique transcriptional reprogramming on the host and fungal side during “prey” digestion. Authors present circumstantial evidence suggesting that fungal proteases might co-function with host protease to promote prey digestion. This is also corroborated by the fact that protease inhibitor treatment abolishes “prey” digestion. Although this might go beyond the scope of this manuscript, authors do not test for causality here and do not provide genetic evidence that fungal genes encoding aspartic proteases are actually contributing to prey digestion. Although I am personally ok with that, the conclusions might remain too speculative to justify for publication in Nature Microbiology.

4) Others

The quality of the supplementary material (especially supplementary figures) could be improved. In many cases, the legends are insufficiently describing the figures and it is difficult to understand what has been done.

Overall, the paper has definitively some potential and is interesting, but I remain skeptical – based on the experimental evidence shown – that the fungus is indeed contributing to prey digestion. Whether this potential plant-fungus co-functioning in prey digestion has actually relevance for plant (and fungal) growth, development and reproductive fitness remains also unclear.

Author Rebuttal to Initial comments

Response to Referees

Reviewer #1 (Remarks to the Author):

In their study by Sun et al aimed to prove the assumption that symbiotic interaction between the sundew *Drosera spatulata* and the acidophilic fungus *Acrodontium crateriforme* is crucial for prey digestion and thus *Drosera*'s carnivorous lifestyle.

We thank the Reviewer for the positive and detailed response.

9To this end, the authors have merged data derived from a set of individual experiments to support above mentioned symbiotic interaction hypothesis.

While the bacterial community found in *Drosera*'s mucilage resembled that of the surrounding plants, the fungal community was significantly different, with *Acrodontium crateriforme* representing the dominant fungal species regardless of sampling site or season. Based on the observation that duration of the digestive process is shortened in traps inoculated with *A. crateriforme*, the authors suggest that the presence of this acidophilic fungus enhanced digestion, thus allowing the plant to effectively consume more prey.

General comment

The reader may want to learn about the microbial situation in *Dionaea*. Thus, the authors may want to cite and discuss:

Venus flytrap carnivorous lifestyle builds on herbivore defense strategies.

Bemm et al. *Genome Res.* 2016 Jun;26(6):812-25.

Venus flytrap microbiotas withstand harsh conditions during prey digestion. Sickel et al. 2019 *FEMS Microbiol Ecol.* Mar 1;95(3). doi: 10.1093/femsec/fiz010. PMID: 30649283.

We were aware of these two classic studies and had previously cited the reviews that referred to them. We have now formally cited the two papers and consolidated the phrases based on these studies.

Questions:

Q1: From their results the authors conclude that *A. crateriforme* enhanced the digestion process of the trap. This could also be interpreted in terms of trap reopening under the control of the that is actually size of food stock present. In other words, when *Drosera* traps and fungus compete for food, the food stock is depleted earlier and thus the trap opens early. A notion that seems supported by the authors own finding `The *A. crateriforme* ammonium transporters as well as

10nitrate reductase also exhibited the same expression trends (Supplementary Fig. 22), suggesting active ammonium exchange and utilisation already taking place within the plant-fungus holobiont.

This finding that the reviewer referred was “For nutrient acquisition related genes, ammonium transporters, nitrate transporter and nitrate reductase in *D. spatulata* showed increased transcript abundance in both digestion and coexistence phases, and are central to nitrogen uptake and assimilation¹⁷ but were more highly expressed in coexistence than digestion phase (determined by DESeq2⁴⁸; $|\log_2$ fold change| > 1 and adjusted P < 0.05; **Supplementary Fig. 19** (L406-L412)” This observation was made during the coexistence phase, in the absence of external food sources, indicating that the ammonium exchange and utilization are likely collaborative interactions between the fungus and the plant, rather than competitive ones. This finding, supported by additional evidence from newly conducted experiments, has been further clarified in the manuscript, with detailed elaboration provided in Discussion L588-596, “This may ultimately result in more digested nutrients to both the fungus and plant host, which was evident from upregulation of sundew’s gene families involved in nitrogen assimilation such as the asparagine synthetases, and transporters that were only up-regulated during prey digestion with *A. crateriforme* present. Together these results also imply that the plant-fungal coexistence is cooperative, and may be mutualistic, as prey capture rates are positively associated with plant fitness⁶⁷ and may be especially relevant in *Drosera* which are carnivorous plants considered sit-and-wait predators.”

Q2: The authors mention ‘The *D. spatulata* DEGs had a higher proportion of multi-function and additive DEGs than the fungus, consistent with the optimisation of the genes repurposed to involve digestion while the majority still retained the ancestral function of species interaction.’ The reader does not quite understand this issue and suggests two perform experiments that compares responses to intact fungus, dead fungus or just plain chitin samples. From this experiment one will learn, whether or not the presence of the elicitors such as chitin are essential and sufficient to trigger *D. spatulata* DEGs, as it is the situation in *Dionaea* (Bemm et al. 2016).

We are grateful to the reviewer, particularly for the suggestion to explore the roles of both intact and deceased *A. crateriforme*, as well as plain chitin, in eliciting transcriptomic responses in *D. spatulata*. Following this advice, our subsequent experiments uncovered that a significant portion of the differentially expressed genes (DEGs), relevant to either the coexistence phase or

both the coexistence and digestion phases, were similarly regulated in the presence of chitin or fungal remnants. However, a smaller subset of DEGs, pertinent to the additive (necessitating the presence of both fungus and prey) or digestion phases, were influenced in sundews treated with either chitin or dead fungus. This distinction elegantly highlights various sets of genes involved in different stages of the digestion process: species interaction/prey detection, prey digestion, and nutrient absorption. Further details of these results are presented in paragraphs **L439-L459**.

This experimental setup not only enriches our manuscript but also bridges the gap in understanding the role of elicitors in gene expression, akin to the mechanisms elucidated in *Dionaea*. We can attribute the overlap of gene sets to the plant priming process, which complements our feeding experiments nicely and provides a more comprehensive overview of the triggers involved in *D. spatulata*'s prey digestion process.

Q3: The authors mention that additional transcriptome sequencing was performed in *D. spatulata* towards the end of digestion process. Concerning this issue the authors may wish to relate their findings to those of a related study on *Dionaea* 'Rentsch et al, 2023. Comparative transcriptomics of Venus flytrap (*Dionaea muscipula*) across stages of prey capture and digestion.'

We appreciate the insights provided by the preprint and have discussed this with the Editor. Additional transcriptome sequencing in two time points of the digestion stage was conducted to i) ensure consistency in stage comparison, and ii) enhance the statistical power of the gene co-expression analysis. Re-analysing and comparing cross-species transcriptomes at different time points necessitates further experimental design and falls outside the scope of our current study.

Q4: The reader is wondering how one can apply "10-5g of sterilized ant powder ...on a single leaf..." (line 686) for the feeding experiments – and appreciates a more detailed description of how to apply such a tiny amount of prey powder in a controlled fashion. The method part should contain information on how the authors assure that the weighed protein amount reaches the proper trap regions based on proper statistics?

We thank the Reviewer for the comment and agree that the methods should be described in more details. We have substantially rewritten the Methods.

In addition to adding minute amounts of prey powder to *Drosera* traps, feeding experiments must be performed with intact living or dead prey with the responses compared.

To elucidate the roles of prey (nutrients), living (mechanical stimuli), and in response to Reviewer 4's comments, we have modified our experimental setup to include mechanical pressures and two non-natural sources: wood powder and shrimp powder. We are confident that the outcomes of these new experiments have further strengthened our findings (L255-L277).

Q5: In lines 271-274, the authors state that “*A. crateriforme* growth is positively correlated with the amount of ant powder to the medium, suggesting that the fungus can utilize insects as growth supplement.”, relating to supplementary Fig. 10. The data shown there, however, is not *A. crateriforme* growth after addition of various amounts of ant powder. Instead, 1g of ant powder is added to three different media. As far as I can see, growth area is smallest (!) in each of the three media containing 1g of ant powder compared to control or dextrose-supplemented medium. So, there's no positive correlation of fungal growth and ant powder supplementation. As already asked above, the authors might therefore want to reconsider the conclusion that the fungus can benefit from insect digestion for its own growth.

We are particularly grateful to the Reviewer for this observation. It was indeed an oversight on our part to conflate results from preliminary tests involving different ant species. Having repeated the experiments, we found that the conclusion remains unchanged. We have now updated the manuscript with the new figure (Supplementary Fig. 6).

Q6: Genome sequencing detected genomic rearrangements in *A. crateriforme* the authors discuss as adaptation to its acidic environment and symbiotic interaction with plants. RNASeq analyses in untreated controls of both *A. crateriforme* and *D. spatulata* and inoculated and/or fed holobiont samples were performed. This complex experiment is difficult for the reader to

understand, partly because of poor description in the supplement (see comment below).

We have thoroughly revised this section of the Results to enhance clarity, particularly concerning the different treatments, which is now elucidated with **Extended Data Fig. 2**.

Q7: As far as the reader understands, data derived from inoculated & fed samples (coexistence & digestion samples as well as holobiont) were mapped to both plant and fungus genomes (lines 718 & 719). Then, the authors state that more than 60% of DEGs “identified in the digestion phase for each species were co-expressed in the coexistence phase” (lines 404-406).

Mapping the reads to both genomes, how can one reliably assign a read to the right species? Obviously, reads mapping to both species’ genomes were removed (lines 720 & 721) - Moreover, the length of the grey “overlap”-bar in Figure 5a doesn’t represent more than 60% of all DEGs, especially for the DEGs downregulated in the fungus. Since this transcriptomic dataset is central for the author’s hypothesis, the reader would like to find the experiment(s) and data analysis properly described.

The percentage actually refers to the DEGs in the digestion phase as the text reads, “58.6–63.8% of differentially expressed genes (DEGs) identified between baseline and digestion phase for each species were also differentially expressed in the coexistence phase in the same trend relative to the baseline (**Fig. 5a**) (**L395-398**)”. We have clarified this previously in the Figure 5a (marked by % of digestion) and now in legend “Percentages indicate the proportion of differentially expressed genes in the digestion phase that were also expressed in the same trend in the coexistence phase. (**L470**)”

On average ~0.01% of reads in the sample were mapped to both genome assemblies and were excluded. Of the reads that were mapped uniquely to one genome, on average 95.6-95.8% and 89-99.2% of the read length in plant-*Acrodontium* samples were mapped to the plant and fungus genome, respectively suggesting these mappings were highly genuine. We have updated these numbers in the Methods and **Supplementary Table S11**.

Q8: To the reader it is not a real surprise that DEGs between digesting and inoculated (coexistence) samples overlap and make the authors speculate about co-option of defense-involved genes for prey digestion. Previous related studies on *Drosera* & other carnivorous

plants before, came to the same conclusion. The author may want to read cite these papers relevant to the topic discussed here.

We have now referenced these studies, making it clear that our research is the first to isolate and investigate sundew digestion responses by inoculating a non-pathogenic fungus.

Q9: The authors focus on genes that were only DEGs in the presence of both fungus and prey (holobiont) and categorize these based on assignment to the individual experiments described before. They suggest that “more than half of upregulated DEGs in both species were designated having multiple or gene-opted”, thus pointing to co-evolution of plant & fungus and optimization of the holobiont transcriptome including synergistic expression of digestive enzymes and nutrient transporters. Finally, the authors quantify JA levels in differently treated traps, pointing to an increase in JA-Ile after “application of both ant powder and microbes” (line 534 & 535). Unfortunately, there’s no *A. crateriforme*-inoculated sample treated with prey to test for an additional boost of JA induction in the presence of the assumed fungal interaction partner.

We have repeated the experiment and quantified the level of JA, this time with *A. crateriforme*-inoculated plants treated with prey. The revised text now reads: “We found that JA levels displayed an increasing trend when plants inoculated with *A. crateriforme* was supplemented with ant powder (**Supplementary Fig. 27**), which correlates with the activation of JA-related priming in plants^{56,57}. (**L528-530**)”. Additionally, we have accordingly revised this paragraph (**L517-535**).

Q10: Coming back to the transcriptomic data, JA-signalling genes are analyzed under various conditions, leading the authors to conclude that JA signalling genes are coexpressed in both inoculated and ant-fed plants (coexistence & digestion). The heatmap (Fig. 6b) shows $\log_2(\text{TPM})$, which makes it hard to see the differences. Many of the genes depicted seem to be comparably expressed under all conditions, even in the control. Maybe giving $(\log_2)\text{FoldChange}$ would bring some more clarity. It is yet unclear, why the authors choose these data for their final figure, since they do not extensively discuss these results. Thus, one would suggest to keep only Figure 6c and move the JA data to the supplement.

We thank the Reviewer for this comment. We have now revised the section accordingly and retained Figure 6c and moved the JA expression to **Supplementary Fig. 27**. The heatmap is now represented in $(\log_2)\text{FoldChange}$.

Minor comments:

- Supplementary Methods & Supplementary figure legends contain numerous typos or are difficult to understand. Example: Supplemental Figure 19: “Coexistence samples mean Acrodontium crateriforme was inoculated on Drosera spatulata for one month. Digestion samples show ant powder add on Drosera spatulata for 3 days. Coexistence digestion samples show ant powder add on inoculated Drosera spatulata for 3 days. To make sure the impact of RNA expression in late stage, we add ant powder add on Drosera spatulata for 107 hours.” The last sentence doesn’t make much sense. The authors may want to carefully revise the supplementary section.

Consequently, we have updated the legends of Supplementary Fig. 19, now presented as **Extended Data Fig. 2**, to reflect these revisions accurately. We have sought assistance from our colleagues and new authors/collaborators with a keen eye for detail to thoroughly review the **Methods and Supplementary Information** of the manuscript.

- Lines 729-731: The authors write “Due to the present (presence??) of peptidase without any expressions across conditions in *D. spatulata*, we removed the 30% lowest-expressed genes in each transcriptome using the sum of samples.” What does that mean? How can you justify omission of data based on intricacy to detect certain transcripts?

It is generally recommended in the building of co-expression networks some genes were removed as correlation measurements are inaccurate in low or no expression genes (“Steps in the network construction include the following: (1) optional filtering of variables (genes) and observations (samples) that contain too many missing values or have zero variance”, the WGCNA documentation <https://cran.r-project.org/web/packages/WGCNA/WGCNA.pdf>). Filtering the transcriptome for improved co-expression network has been recently discussed (Sanchez-Baizan *et al.*, 2022 BMC Biology), and we followed the approach recommended by the WGCNA authors. We have made this process clearer in the Methods section as “Prior to

construction of the network, the 30% lowest-expressed genes in each transcriptome using the sum of samples were excluded. (L757-759)”

Reviewer #2 (Remarks to the Author):

This paper identifies *Acrodontium crateriforme* as the dominant fungal constituent of the *Drosera* mucilage engaged in insect capture and digestion. The data nicely demonstrate the role of the fungus in insect and protein digestion and implicate it as the major partner allowing the evolution of insectivory.

We thank the reviewer for the comment.

Reviewer #3 (Remarks to the Author):

This manuscript describes a very interesting association between a fungus and the carnivorous sundew plant. This work is an excellent investigation and presents a series of future questions to base on using these genomic and analytic analyses.

Overall I find this a well written and analyzed project. The amplicon, culturing, metabolism and comparative analysis provide a clear perspective on this fungal-plant interaction. I find the work fascinating and very interesting exploration of plant-fungal interactions in a really quite different dimension. I appreciate the efforts as well to connect the mycobiome profile to those of pitcher plant work (*Nepenthes* sp) as I would have been curious or expected there to be some comparative analyses done - this is a good context to have in this manuscript.

While not critical -- I do not think the authors have space to add more analyses - but I wonder if the authors had looked at the MATing type locus for its presence in the assembled genome and if the architecture give you any ideas on mating capabilities?

Following the approach of Aylward et. al, *A. crateriforme* is likely heterothallic which is similar to sister species *Neohortaea acidophila* and *Baudoinia panamericana*. We have added this information in the manuscript as the following text,

17“The mating locus and its adjacent orthologs of *A. crateriforme* were determined and this was in syntenic with sister species (**Supplementary Fig. 11**), suggesting that this fungus is likely heterothallic, similar to the ancestral Mycosphaerellales³⁷ (**L325-L328**)”

Around line 370 there is discussion of chromosomal synteny but some local synteny which was coined as 'mesosynteny' in Hane et al 2011 <https://link.springer.com/article/10.1186/gb-2011-12-5-r45> -- so not sure if that more general concept is what applies here or if the Capnodiales pattern is distinct as cited for Ohm et al 2012.

We agree that what we observed is indeed mesosynteny and have updated as the text below:

“Gene order within linkage groups has been lost (**Supplementary Fig. 16**), suggesting extensive intra-chromosomal rearrangements, which appear to be a hallmark of mesosynteny observed in various fungal taxa⁴³ (**L360-L363**)”

Note that *Hortaea acidophila* is now named *Neohortaea acidophila* <https://ncbi.nlm.nih.gov/Taxonomy/Browser/wwwtax.cgi?id=245834> <https://doi.org/10.3767/003158514X681981> so this should be updated in the trees.

We have updated the species name in the text and Figures accordingly.

Reviewer #4 (Remarks to the Author):

The manuscript by Sun and colleagues is describing a potentially interesting finding, namely that prey digestion in a carnivorous plant is promoted by the presence of an acidophilic fungus.

18The manuscript has potential but there are numerous important issues that hinder the potential impact of the work.

We thank the reviewer for the comment. We have added various experiments which believe

1) Microbiome profiling

Authors showed interesting microbiome profiling data and convincingly showed that the mucilage microbiome has a distinct fungal microbiome. However, they used the very outdated OTU-based clustering method. Nothing against this but authors might have underestimated the true diversity of bacteria and fungi in their samples, including in mucilage.

We concur with the reviewer that the ASV (Amplicon Sequence Variant) approach has been standardized in amplicon analyses, particularly for bacterial 16S datasets. However, we chose to employ the OTU (Operational Taxonomic Unit) approach for two reasons. Firstly, our positive controls using fungal mock communities indicated that the USEARCH pipeline tends to overestimate the number of ASVs, likely due to insufficient sequence correction—a situation that can be corrected by clustering sequences into OTUs. Secondly, we identified four ASVs classified as *A. crateriforme* that were 99.98% similar to each other, differing by only two nucleotides. Given our focus on the fungus-plant relationship is at the species level, we opted for the OTU approach to report our observations. For the reviewer's information, both the ASV and OTU methodologies led to the same conclusions.

My major concern is more that the samples were collected from a geographically-narrow area and it remains unclear whether this association remains stable across larger geographical scales. It would be important to test whether *D. spatulata* host similar or dissimilar dominant fungal taxa in at least one completely independent plant population outside Taiwan. Authors actually showed that *A. crateriforme* might not be a dominant microbiota member in other carnivorous plants. In other words, is this potentially beneficial association retained, irrespective of the environment? or can different fungi functionally complement *A. crateriforme* in other

19environments to promote prey digestion? This information would elevate the potential impact of the work.

We are excited to share with the reviewer that we now have amplicon data from both the USA and the UK. Our analysis revealed that *A. crateriforme* is present in multiple *Drosera* species and has been frequently observed as the dominant species globally. We agree with the reviewer and believe that this finding significantly elevates the potential impact of our work.

2) Experimental evidence

The experimental evidence demonstrating that the fungus is needed for efficient prey digestion **is rather weak** (Figure 3e,f) and I personally think that these experiments are insufficient to conclude that the fungus is integral to prey digestion. In these experiments, authors prepared sterile carnivorous plants, re-colonized these plants with the fungus and test whether fungal inoculation leads to change in tentacle behavior or promote biotin-labelled BSA breakdown. The proxy used to determine prey digestion is based on opening and closing time of tentacles. Why using ant powder and why looking at such indirect proxy? Authors might consider a negative control such as inert powder or powder from another (less digestible, non insect) organism that is not a natural sundew's prey. Why not setting up experiments with dead ants and monitor ant biomass degradation based on quantitative measurements? A potential issue is that the presence of the fungus might also modulate tentacle behavior independently from prey digestion.

The experiment with BSA is important because it truly reflects protein degradation capability rather than tentacle behavior. However, one need to realize that the fungal effect appears rather limited in this case. Can the same weak effect be observed with other non-pathogenic fungi? Does this effect depend on the concentration of the fungal inoculum?

We are grateful for the reviewer's feedback on the experimental evidence. Acting on the reviewer's advice, we redesigned the experiment to include additional substrates such as ground shrimp shells and wood powders, which serve as controls not representative of sundew prey. Additionally, we recolonised sterile plants with microbiota from the mucilage of wild sundews and the leaf surfaces of co-occurring plants.

The revised experiments offered fresh insights, revealing that i) inoculating microorganisms lead to a quicker reopening of traps, ii) specifically inoculating with *A. crateriforme* results in the fastest trap reopening, and iii) significant degradation of Bovine Serum Albumin (BSA) was consistently observed only in the mucilage collected from sundews inoculated with *A. crateriforme*, unlike those collected from sterile plants or plants inoculated from natural microbiota.

Coupled with new RNA-seq data of sterile sundews treated with chitin or dead fungus (as suggested by Reviewer one), it appears that exposing plants to microorganisms results in a primed state of enhanced defence and, in the case of sundews, leads to a more efficient trapping process. Furthermore, *A. crateriforme* has shown to enhance the digestion process through its enhanced protein degradation capability.

Variations were observed between batches in the BSA degradation experiments, although the overall trend remained consistent. In response, we elected to perform statistical analyses between batches and have presented the data as independent batches (**Fig. 3e** and **Supplementary Fig. 10**).

The microscopic pictures showed very different colonization patterns between *A. crateriforme* recolonized plants and natural plants, raising the possibility that in nature, *A. crateriforme* is only one, out of many others that potentially contribute to prey digestion in mucilage. Authors could also consider microbiota transplantation from natural plants to sterile plants and test whether prey digestion with *D. spatulata* alone recapitulates prey digestion of plants recolonized with more complex mucilage microbiomes. Is *D. spatulata* really the major player in this process?

Please see our detailed previous comment.

The number of replicates (n=5 technical replicates) for these experiments is also questionable. Based on what I can read, these experiments were not even performed three times independently but derived from one experiment with 5 technical replicated. For publication in

21Nature Microbiology, fully independent replicates should be presented.

The experiments were conducted with biological replicates. To mitigate batch effects, we have repeated the experiments incorporating new variables across five biological replicates, This approach and the detailed procedure have been clearly articulated in the Methods section “**Feeding experiment of *D. spatulata* containing different microbiota**”: “Each replicate was a single leaf from a sundew and each comparison consists of five leaves from five independent sundew plants. (L699-701)”

3) RNA-Seq

Authors nicely showed that there is a unique transcriptional reprogramming on the host and fungal side during “prey” digestion. Authors present circumstantial evidence suggesting that fungal proteases might co-function with host protease to promote prey digestion. This is also corroborated by the fact that protease inhibitor treatment abolishes “prey” digestion. Although this might go beyond the scope of this manuscript, authors do not test for causality here and do not provide genetic evidence that fungal genes encoding aspartic proteases are actually contributing to prey digestion. Although I am personally ok with that, the conclusions might remain too speculative to justify for publication in Nature Microbiology.

We are trying to establish the genetic and molecular tools in *A. crateriforme* but so far have been challenging in this group of extremophilic fungi. We believe the comparison between trap opening and BSA degradation performed on new substrates and different sources of transplantations enable a more thorough discussion. We have updated the text accordingly.

4) Others

The quality of the supplementary material (especially supplementary figures) could be improved. In many cases, the legends are insufficiently describing the figures and it is difficult to understand what has been done.

We have sought assistance from our colleagues and new authors/collaborators with a keen eye for detail to thoroughly review the Supplementary Information of the manuscript.

Overall, the paper has definitively some potential and is interesting, but I remain skeptical – based on the experimental evidence shown – that the fungus is indeed contributing to prey digestion. Whether this potential plant-fungus co-functioning in prey digestion has actually relevance for plant (and fungal) growth, development and reproductive fitness remains also unclear.

We extend our gratitude to the Reviewer for their comment acknowledging the interest of our study. We are convinced that confirming the presence and dominance of *A. crateriforme* across multiple *Drosera* species globally, along with the additional results from microbiota transplantation experiments suggested by the Reviewer, demonstrates that *A. crateriforme* alone can indeed "recapitulate prey digestion in plants recolonized with more complex mucilage microbiomes." This finding significantly contributes to our understanding of prey digestion processes in carnivorous plants.

Decision Letter, first revision:

Message: Our ref: NMICROBIOL-23102735A

17th May 2024

Dear Jason,

Thank you for submitting your revised manuscript "An acidophilic fungus is integral to prey digestion in a carnivorous plant" (NMICROBIOL-23102735A). It has now been seen by three of the original referees and their comments are below. The reviewers find that the paper has improved in revision, and therefore we'll be happy in principle to publish it in *Nature Microbiology*, pending minor revisions to satisfy the referees' final requests and to comply with our editorial and formatting guidelines.

23We are now performing detailed checks on your paper and will send you a checklist detailing our editorial and formatting requirements in about a week. Please do not upload the final materials and make any revisions until you receive this additional information from us.

Thank you again for your interest in Nature Microbiology Please do not hesitate to contact me if you have any questions.

Sincerely,

Reviewer #1 (Remarks to the Author):

The authors properly answered my questions but Q3.

By addressing my questions the paper gained a lot.

Reviewer #3 (Remarks to the Author):

I've read the response and revision and believe the authors have addressed to the best of their ability the queries and concerns. I think this revision has managed the concerns I raised as a reviewer and most of those described by the others.

Reviewer #4 (Remarks to the Author):

The authors considered very carefully all the concerns and performed the necessary additional experiments to provide data in support to all statements. I congratulate the authors for this very interesting study, which is now, in my opinion, acceptable for publication in Nature Microbiology.

Author Rebuttal, first revision:

Response to Referees

Reviewer #1: Remarks to the Author:

The authors properly answered my questions but Q3.

24By addressing my questions the paper gained a lot.

Reviewer #3: Remarks to the Author:

I've read the response and revision and believe the authors have addressed to the best of their ability the queries and concerns. I think this revision has managed the concerns I raised as a reviewer and most of those described by the others.

Reviewer #4: Remarks to the Author:

The authors considered very carefully all the concerns and performed the necessary additional experiments to provide data in support to all statements. I congratulate the authors for this very interesting study, which is now, in my opinion, acceptable for publication in Nature Microbiology.

We thank all three reviewers for the positive approval to the revised manuscript.

Final Decision Letter:

Message 19th June 2024

:

Dear Jason,

I am pleased to accept your Article "An acidophilic fungus promotes prey digestion in a carnivorous plant" for publication in Nature Microbiology. Thank you for having chosen to submit your work to us and many congratulations.

You may wish to make your media relations office aware of your accepted publication, in case they consider it appropriate to organize some internal or external publicity. Once your paper has been scheduled you will receive an email confirming the publication details. This

25is normally 3-4 working days in advance of publication. If you need additional notice of the date and time of publication, please let the production team know when you receive the proof of your article to ensure there is sufficient time to coordinate. Further information on our embargo policies can be found here:

<https://www.nature.com/authors/policies/embargo.html>

Please note that *Nature Microbiology* is a Transformative Journal (TJ). Authors may publish their research with us through the traditional subscription access route or make their paper immediately open access through payment of an article-processing charge (APC). Authors will not be required to make a final decision about access to their article until it has been accepted. Find out more about Transformative Journals

Congratulations once again and I look forward to seeing the article published.

With kind regards,